# Scalable Signature Kernel Computations via Local Neumann Series Expansions

**Matthew Tamayo-Rios**
Seminar for Applied Mathematics & AI Center
ETH Zurich, Switzerland

**Alexander Schell**
Department of Mathematics
Technical University of Munich, Germany

**Rima Alaifari**
Department of Mathematics
RWTH Aachen University, Germany

## Abstract

The signature kernel [10] is a recent state-of-the-art tool for analyzing high-dimensional sequential data, valued for its theoretical guarantees and strong empirical performance. In this paper, we present a novel method for efficiently computing the signature kernel of long, high-dimensional time series via adaptively truncated recursive local power series expansions. Building on the characterization of the signature kernel as the solution of a Goursat PDE [17], our approach employs tilewise Neumann-series expansions to derive rapidly converging power series approximations of the signature kernel that are locally defined on subdomains and propagated iteratively across the entire domain of the Goursat solution by exploiting the geometry of the time series. Algorithmically, this involves solving a system of interdependent Goursat PDEs via adaptively truncated local power series expansions and recursive propagation of boundary conditions along a directed graph in a topological ordering. This method strikes an effective balance between computational cost and accuracy, achieving substantial performance improvements over state-of-the-art approaches for computing the signature kernel. It offers (a) adjustable and superior accuracy, even for time series with very high roughness; (b) drastically reduced memory requirements; and (c) scalability to efficiently handle very long time series (one million data points or more) on a single GPU. As demonstrated in our benchmarks, these advantages make our method particularly well-suited for rough-path-assisted machine learning, financial modeling, and signal processing applications involving very long and highly volatile sequential data.

## 1  Introduction

Time series data is ubiquitous in contemporary data science and machine learning, appearing in diverse applications such as satellite communication, radio astronomy, health monitoring, climate analysis, and language or video processing, among many others [21]. The sequential nature of this data presents unique challenges, as it is characterised by temporal dependencies and resulting structural patterns that must be captured efficiently to model and predict time-dependent systems and phenomena with accuracy. Robust and scalable tools for handling such data in their full temporal complexity are thus essential for advancing machine learning applications across these domains. One particularly powerful approach in this direction is the *signature kernel* [10, 17], the Gram matrix of a high-fidelity feature embedding rooted in rough path theory and stochastic analysis [14, 8], which has gained relevance as an increasingly popular tool in the modern analysis of sequential data [11, 13].

39th Conference on Neural Information Processing Systems (NeurIPS 2025).

Conceptually, the signature kernel (of a family of multidimensional time series) is the Gram matrix of the *signature transform*, a highly informative and faithful feature map that embeds time series into a Hilbert space via their iterated integrals, thus uniquely capturing their geometry and essential time-global characteristics in a hierarchical manner [9]. This structure situates the signature kernel naturally within the framework of reproducing kernel Hilbert spaces (RKHS) and enables its practical utility through kernel learning techniques. The rich intrinsic properties of the underlying signature transform further confer several strong theoretical properties to the signature kernel, including invariance under irregular sampling and reparametrization, universality and characteristicness, and robustness to noise [6]. Its wide-ranging theoretical interpretability combined with its strong real-world efficiency have elevated the signature kernel to a state-of-the-art tool for the analysis of time-dependent data [13].

However, existing methods for computing the signature kernel suffer from significant scalability issues, particularly when dealing with long or highly variable time series.[1] The reason is that these methods typically either approximate truncations of the signature transforms via dynamic programming [10] or solve a global Goursat PDE for the signature kernel using two-dimensional FDM discretizations [17], which both involve at least quadratic storage complexity relative to the length $\ell$ of the time series, thus resulting in prohibitive memory usage for long or very rough time series. These limitations, and the quadratic complexity in the number of constituent time series (a common kernel method bottleneck not specific to the signature kernel), can severely obstruct the application of signature kernels to large-scale, real-world datasets; see, e.g., [19, 11] and the references therein.

To address this, we introduce a novel approach to compute the signature kernel based on adaptively truncated local (tilewise) Neumann series expansions of the solution to its characterizing Goursat PDE; see Figure 1 for an overview. Partitioning the domain $[0, 1]^2$ of this PDE solution into tiles according to the data points of the input time series, we derive rapidly convergent power series representations of the solution on each tile, with boundary data obtained from adjacent tiles. This enables efficient computation of the signature kernel in terms of memory and runtime, while ensuring superior accuracy and significantly improved scalability with respect to the length and roughness of the kernel's constituent time series. Since only local expansions are stored—rather than a global $\ell \times \ell$ grid—memory usage grows much more slowly with $\ell$, allowing the method to handle very long time series, even up to a million points each on a single GPU, where PDE or dynamic-programming-based solvers typically run out of memory. By explicitly leveraging the time-domain geometry of piecewise-linear time series in this way, our method supports localized computations of the signature kernel that are both highly parallelizable and memory-efficient, even for very long and rough time series.

Specifically, our main contributions are:

1. **Neumann Series for the Signature Kernel:** We propose a tilewise integral-equation approach to construct recursive local power series expansions of the signature kernel, based on boundary-to-boundary propagation and paired with an adaptive truncation strategy. This leverages the kernel's time-domain geometry to enable its computation with significantly reduced memory requirements and computational cost without compromising accuracy.

2. **Parallelizable Local Computation:** Exploiting the piecewise-linear structure of the input time series, we partition the full kernel domain into ordered tiles supporting parallel local Neumann expansions with adjustable precision and minimal global communication.

3. **Scalability to Very Long and Rough Time Series:** Our method, termed `PowerSig`, achieves scalability to very long (over $10^6$ points on a single GPU) and highly volatile (rough) time series, addressing key limitations of existing methods.

4. **Empirical Validation On Benchmarks:** We demonstrate the practical advantages of our method through comprehensive benchmarks against several state-of-the-art signature kernel solvers, demonstrating superior accuracy, runtime, and memory efficiency of our approach.

The remainder of this paper elaborates on these contributions, starting with a brief review of related work (Section 1.1), followed by a detailed description of our methodology (Section 2) and a presentation of our experimental results (Section 3). Proofs are given in Appendix A.

---

[1] Variability of a (discrete-time) time series is quantified by the sum of squared differences between consecutive points of the time series and referred to as its 'roughness'.

The codebase for our method, including all implementation details, is provided in the supplementary material and publicly available at: `https://github.com/geekbeast/powersig`.

## 1.1 Related Work

The computation of Gram matrices for signature-transformed time series was first systematically studied by Király and Oberhauser [10], who identified the signature kernel as a foundational link between rough path theory, data science, and machine learning—an interplay broadly envisioned earlier by Lyons [12]. Chevyrev and Oberhauser [6] further extended these theoretical foundations, and introduced a statistically robust variant of the signature kernel via appropriate time series scaling. Computationally, Salvi et al. [17] significantly advanced beyond earlier dynamic programming approaches for computing truncated kernels [10] by characterizing the (untruncated) signature kernel through a linear second-order hyperbolic (Goursat) PDE. This approach, implemented in widely-used libraries such as `sigkernel` and GPU-accelerated `KSig` [19], provided efficient signature kernel computation through finite-difference PDE approximations.

However, PDE-based methods, despite providing highly parallelizable and fast routines for short-to moderate-length time series, exhibit poor scalability due to quadratic memory usage, becoming impractical for long or rough time series beyond a few thousand to tens of thousands of time steps [19, 11] (our experiments confirm computational limits of approximately $10^3$ steps for `sigkernel` and $16 \times 10^4$ for `KSig` on 4090 RTX GPUs). Dynamic programming remains an option for computing kernels of truncated signatures [10], although it likewise suffers from poor scalability. Recent efforts to reduce this cost through random Fourier features or other low-rank approximations [20] often degrade in accuracy for larger time-series length or time series of higher dimension.

Our proposed method, `PowerSig`, circumvents these issues through tilewise local power series expansions of the signature kernel, avoiding the need for a global storage footprint that scales quadratically with time series length. By viewing the Goursat PDE as a Volterra integral equation, we can compute rapidly convergent Neumann series expansions of the kernel locally, storing only series coefficients rather than full two-dimensional arrays. This localized strategy enables fast and accurate signature kernel computations for extremely long (over $10^6$ points) and high-dimensional time series on single GPUs, significantly improving scalability and efficiency.

During the preparation of this manuscript, we came across a preprint by Cass et al. [5] which appeared concurrently to the release of our first version. Their work is also based on recursive power series expansions of the signature kernel, albeit on different mathematical and algorithmic premises. An empirical and conceptual comparison between our method and theirs is provided in Section B.2.

## 2 Signature Kernels via Recursive Local Neumann Series

This section presents our tilewise Neumann-series expansion method for the signature kernel. We begin by revisiting the PDE characterization of the signature kernel [17] (Section 2.1) and recast it as an equivalent integral equation that allows for a recursive, tile-based decomposition of its solution with only boundary values exchanged between adjacent tiles (Section 2.2). This local formulation produces rapidly convergent nested power series expansions on each tile and supports an adaptive truncation scheme to balance accuracy and computational cost (Section 2.3). By storing and passing only local series coefficients, the method achieves substantial memory savings compared to global PDE or dynamic-programming solvers (Section 3). Figure 1 summarizes the core idea.

## 2.1 The Signature Kernel

Let $\boldsymbol{x} = (\boldsymbol{x}_1, \ldots, \boldsymbol{x}_\ell) \subset \mathbb{R}^d$ and $\boldsymbol{y} = (\boldsymbol{y}_1, \ldots, \boldsymbol{y}_\ell) \subset \mathbb{R}^d$ be time series of common length $\ell \in \mathbb{N}$ (if lengths differ, one may pad the shorter time series by repeating its final entry). For any such time series $\boldsymbol{z} := (\boldsymbol{z}_1, \ldots, \boldsymbol{z}_\ell)$, define its *affine interpolant* $\hat{\boldsymbol{z}} : [0, 1] \to \mathbb{R}^d$ by

$$\hat{\boldsymbol{z}}(t) := \boldsymbol{z}_k + (\ell - 1)\left(t - \tfrac{k-1}{\ell-1}\right)\Delta_k \boldsymbol{z}, \quad t \in \left[\tfrac{k-1}{\ell-1}, \tfrac{k}{\ell-1}\right], \quad (k = 1, \ldots, \ell - 1),$$

where $\Delta_k \boldsymbol{z} := \boldsymbol{z}_{k+1} - \boldsymbol{z}_k$. (This is just the unique continuous piecewise linear function interpolating the points $\boldsymbol{z}_1, \ldots, \boldsymbol{z}_\ell$.) The derivative of $\hat{\boldsymbol{z}}$ (defined almost everywhere on $[0, 1]$) is given by

$$\hat{\boldsymbol{z}}'(t) := (\ell - 1)\Delta_k \boldsymbol{z}, \quad t \in \left(\tfrac{k-1}{\ell-1}, \tfrac{k}{\ell-1}\right).$$

For later use, we now partition the unit square $[0, 1]^2$ into *tiles* as follows. Set

$$\sigma_k := \frac{k-1}{\ell-1} \quad \text{and} \quad \tau_l := \frac{l-1}{\ell-1} \quad (k, l = 1, \ldots, \ell),$$

and define the open tiles and their closures (in the Euclidean topology on $\mathbb{R}^2$) as

$$\mathcal{D}_{k,l} := (\sigma_k, \sigma_{k+1}) \times (\tau_l, \tau_{l+1}) \quad \text{and} \quad T_{k,l} := \overline{\mathcal{D}_{k,l}} = [\sigma_k, \sigma_{k+1}] \times [\tau_l, \tau_{l+1}]. \tag{1}$$

The tiled domain is then given by $\mathcal{D} := \bigcup_{k,l=1}^{\ell-1} \mathcal{D}_{k,l}$, with boundary $\partial \mathcal{D} = [0, 1]^2 \setminus \mathcal{D}$.

Finally, let $\langle \cdot, \cdot \rangle$ denote the Euclidean inner product on $\mathbb{R}^d$. Building on the PDE characterization in [17, Theorem 2.5], we can[2] introduce our main object of interest as follows.

**Definition 2.1** (Signature Kernel). The *signature kernel* of $\boldsymbol{x}$ and $\boldsymbol{y}$ is the unique continuous function $K \equiv K_{\boldsymbol{x},\boldsymbol{y}} : [0, 1]^2 \to \mathbb{R}$ solving the hyperbolic (Goursat) boundary value problem

$$\begin{cases} \dfrac{\partial^2 K(s,t)}{\partial s \partial t} = \rho_{\boldsymbol{x},\boldsymbol{y}}(s,t) K(s,t), & (s,t) \in \mathcal{D}, \\ K(0, \cdot) = K(\cdot, 0) = 1, \end{cases} \tag{2}$$

where the coefficient function $\rho_{\boldsymbol{x},\boldsymbol{y}} : \mathcal{D} \to \mathbb{R}$ is defined tilewise by

$$\rho_{\boldsymbol{x},\boldsymbol{y}}(s,t) := \langle \hat{\boldsymbol{x}}'(s), \hat{\boldsymbol{y}}'(t) \rangle, \quad (s,t) \in \mathcal{D}_{k,l}.$$

An equivalent formulation defines $\rho_{\boldsymbol{x},\boldsymbol{y}}$ on all of $[0, 1]^2$ via

$$\rho_{\boldsymbol{x},\boldsymbol{y}} : [0, 1]^2 \ni (s,t) \mapsto (\ell-1)^2 \sum_{k,l=1}^{\ell} \rho_{k,l} \mathbb{1}_{\mathcal{D}_{k,l}}(s,t), \quad \rho_{k,l} := \Delta_l \boldsymbol{x} \cdot \Delta_k \boldsymbol{y}. \tag{3}$$

In integral form, the boundary value problem (2) is then equivalent to the Volterra integral equation

$$K(s,t) = 1 + \int_0^t \int_0^s \rho_{\boldsymbol{x},\boldsymbol{y}}(u,v) \, K(u,v) \, \mathrm{d}u \, \mathrm{d}v, \quad (s,t) \in [0, 1]^2. \tag{4}$$

Standard fixed-point arguments (Appendix A.2) guarantee that (4) has a unique solution in $C([0, 1]^2)$, the space of continuous functions on $[0, 1]^2$. This yields the following well-known result:

**Proposition 2.2.** *The Goursat problem* (2) *has a unique solution in* $C([0, 1]^2)$*; in particular, the signature kernel of $\boldsymbol{x}$ and $\boldsymbol{y}$ is well-defined.*

The advantage of the Volterra formulation (4) is that it naturally leads to a recursive power series expansion for the signature kernel $K$, as we will now explain. A key step towards this observation, which is also used in the proof of Proposition 2.2, is the following lemma:

**Lemma 2.3.** *For any bounded, measurable function $\varrho : I \to \mathbb{R}$ defined on a closed rectangle $I \equiv [a_1, b_1] \times [a_2, b_2] \subseteq [0, 1]^2$, the integral operator $\boldsymbol{T}_\varrho : (C(I), \|\cdot\|_\infty) \to (C(I), \|\cdot\|_\infty)$ given by*

$$\boldsymbol{T}_\varrho f(s,t) = \int_{a_2}^t \int_{a_1}^s \varrho(u,v) \, f(u,v) \, \mathrm{d}u \, \mathrm{d}v, \quad (s,t) \in I,$$

*has spectral radius zero, that is, $r(\boldsymbol{T}_\varrho) := \sup_{\lambda \in \sigma(\boldsymbol{T}_\varrho)} |\lambda| = 0$. Here, $C(I)$ denotes the space of continuous functions on $I$, equipped with the supremum norm $\|f\|_\infty := \sup_{(s,t) \in I} |f(s,t)|$.*

This fact will justify the use of Neumann series when inverting operators of the form $\mathrm{id} - \boldsymbol{T}_\varrho$ below.

## 2.2 A Recursive Local Power Series Expansion of the Signature Kernel

Our computational strategy is to approximate the signature kernel $K$ by a directed family of local power series expansions constructed recursively over the tiles (1). In the spirit of the Adomian Decomposition Method (ADM) [1, 2, 22, 3, 4], we seek to establish a representation

$$K(s,t) = \sum_{i,j=0}^{\infty} K_{i,j}(s,t) \tag{5}$$

---

[2]Introducing the signature kernel of $\boldsymbol{x}$ and $\boldsymbol{y}$ as the solution to its characterizing PDE circumvents the need to specify it as the inner product of the signature transforms of $\boldsymbol{x}$ and $\boldsymbol{y}$, as originally formulated in [10, 17].

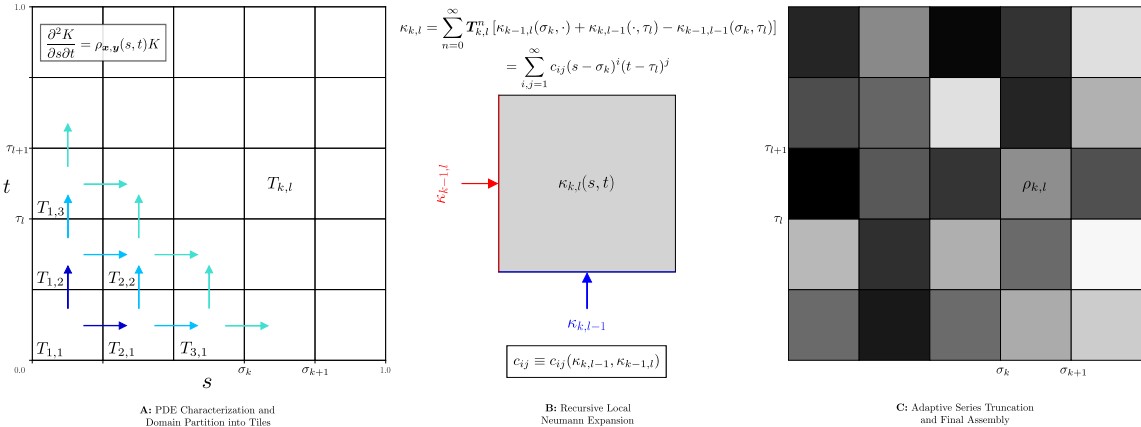

**A:** PDE Characterization and Domain Partition into Tiles

**B:** Recursive Local Neumann Expansion

**C:** Adaptive Series Truncation and Final Assembly

Figure 1: *Summary of our method (*`PowerSig`*) for computing the signature kernel of two time series via recursive local Neumann expansions.* **Panel A**: *The PDE $\frac{\partial^2 K}{\partial s \partial t} = \rho_{\boldsymbol{x},\boldsymbol{y}} K$ induces an $(\boldsymbol{x},\boldsymbol{y})$-dependent partition of $[0,1]^2$ into tiles $T_{k,l}$. Arrows indicate the sequential propagation of boundary conditions across tiles, with decreasing colour intensity corresponding to later propagation steps. Tiles receiving arrows of the same colour form groups whose local series can be computed in parallel.* **Panel B**: *On each given tile $T_{k,l}$, the kernel admits the recursive local Neumann series expansion: $\kappa_{k,l}(s,t) = \sum_{n=0}^{\infty} T_{k,l}^n [\kappa_{k-1,l}(\sigma_k,\cdot) + \kappa_{k,l-1}(\cdot,\tau_l) - \kappa_{k-1,l-1}(\sigma_k,\tau_l)] = \sum_{i,j=1}^{\infty} c_{ij}(s-\sigma_k)^i(t-\tau_l)^j$, which converges uniformly on $(s,t) \in T_{k,l}$. These tilewise expansions depend on boundary values from neighbouring tiles ($\kappa_{k-1,l}$ and $\kappa_{k,l-1}$), with arrows indicating the directions of integration from boundaries to the tile interior.* **Panel C** *illustrates adaptive series truncation and final kernel assembly. Tile shading intensity encodes local truncation depth, which is adaptively determined by the magnitude of $\rho_{k,l} \equiv (\boldsymbol{x}_{k+1} - \boldsymbol{x}_k)(\boldsymbol{y}_{l+1} - \boldsymbol{y}_l)$. Darker tiles indicate the necessity for deeper (higher-order) expansions, while lighter tiles allow shallower truncation.*

with terms $K_{i,j} : [0,1]^2 \to \mathbb{R}$ that are easy to compute and, for our purposes, take the form

$$K_{i,j}(s,t) = \sum_{k,l=1}^{\ell-1} \mathbb{1}_{\hat{T}_{k,l}}(s,t)\, c_{k,l}^{(i,j)}\, (s-\sigma_k)^i(t-\tau_l)^j \tag{6}$$

with some tile-dependent coefficient sequences

$$c_{k,l} \coloneqq \big(c_{k,l}^{(i,j)} \,\big|\, (i,j) \in \mathbb{N}_0^2\big) \in \ell_1(\mathbb{N}_0^2), \tag{7}$$

which are summable and (for $(k,l) \neq (1,1)$) defined recursively by

$$c_{k,l} = \phi_{k,l}\big(c_{k,l-1}, c_{k-1,l}\big) \quad \text{for maps} \quad \phi_{k,l} : \ell_1(\mathbb{N}_0^2)^{\times 2} \to \ell_1(\mathbb{N}_0^2) \tag{8}$$

where $\phi_{k,1}$ (resp. $\phi_{1,l}$) depends only on its second (resp. first) argument.

The sets $\hat{T}_{k,l}$ are half-open, $\hat{T}_{k,l} \coloneqq \big[\sigma_k, \sigma_{k+1}\big) \times \big[\tau_l, \tau_{l+1}\big)$ for $k,l < \ell-1$, and closed on the last row/column (i.e. when $k = \ell-1$ or $l = \ell-1$), $\hat{T}_{k,\ell-1} \coloneqq T_{k,\ell-1}, \hat{T}_{\ell-1,l} \coloneqq T_{\ell-1,l}$. Thus $(\hat{T}_{k,l})_{k,l=1}^{\ell-1}$ defines a partition of the Goursat domain $[0,1]^2$.

We organize the recursion (8) over all $(\ell-1)^2$ tiles, computing the coefficients (7) on each tile $T_{k,l}$ via a Neumann expansion from boundary data on the adjacent tiles $T_{k-1,l}, T_{k,l-1}$ (whose coefficients ($c_{k-1,l}$ and $c_{k,l-1}$) are already available). Section 2.2.2 details the procedure, beginning with the bottom-left tile $T_{1,1}$ in Section 2.2.1.

The goals of this scheme are twofold: (a) to choose the coefficients (7) so that the series (6) converge rapidly on each tile, giving a power series representation of the signature kernel localizations

$$\kappa_{k,l} \coloneqq K|_{T_{k,l}} \,:\, [\sigma_k, \sigma_{k+1}] \times [\tau_l, \tau_{l+1}] \ni (s,t) \longmapsto K(s,t) \in \mathbb{R}, \tag{9}$$

and (b) to truncate these tilewise series expansions (9) so as to obtain a numerically stable and efficient global approximation of the whole kernel $K$; see Section 2.3 for both.

### 2.2.1 Rapidly Convergent Power Series on the First Tile

On the first tile $T_{1,1} = [0, \sigma_2] \times [0, \tau_2]$, we adopt (5)–(7) as an ansatz and assume[3]

$$\kappa_{1,1}(s,t) = \sum_{i,j=0}^{\infty} K_{i,j}(s,t), \quad \text{where} \quad K_{i,j}(s,t) = c_{1,1}^{(i,j)} s^i t^j \tag{10}$$

with $\sum_{i,j} |c_{1,1}^{(i,j)}| < \infty$ and only diagonal coefficients nonzero ($c_{1,1}^{(i,j)} = 0$ for $i \neq j$). Then $\int_{T_{1,1}} \sum_{i,j} |K_{i,j}(w)| \, \mathrm{d}w < \infty$, and Fubini applied to the integral equation (4) gives

$$\sum_{i,j=0}^{\infty} K_{i,j} = \sum_{i,j=0}^{\infty} \tilde{K}_{i,j} \quad \text{with} \quad \tilde{K}_{i,j}(s,t) := \int_0^t \int_0^s \rho_{\boldsymbol{x},\boldsymbol{y}}(u,v) \, K_{i-1,j-1}(u,v) \, \mathrm{d}u \, \mathrm{d}v \tag{11}$$

and for $\tilde{K}_{0,0} \equiv 1$ pointwise on $T_{1,1}$. Since $\rho_{\boldsymbol{x},\boldsymbol{y}}|_{T_{1,1}} \equiv \rho_{1,1}$ (see (3)), a simple induction yields

$$\tilde{K}_{i,j}(s,t) = \begin{cases} \frac{\rho_{1,1}^i}{(i!)^2} \cdot s^i t^j & \text{if } i = j, \\ 0 & \text{if } i \neq j, \end{cases} \quad \text{pointwise on } T_{1,1}.$$

By the identity theorem for power series, (11) implies that the desired $(c_{1,1}^{(i,j)} \mid i, j \in \mathbb{N}_0)$ must read

$$c_{1,1}^{(i,j)} = \frac{\rho_{1,1}^i}{(i!)^2} \cdot \delta_{i,j}, \tag{12}$$

where $\delta_{i,j}$ is the Kronecker delta. On $T_{1,1}$, the decomposition ansatz (10) thus yields the well-known

**Lemma 2.4.** *On the first tile $T_{1,1}$, the signature kernel* (4) *has the form:*

$$K(s,t) = \sum_{i=0}^{\infty} \frac{\rho_{1,1}^i}{(i!)^2} s^i t^i = \begin{cases} J_0\left(2\sqrt{|\rho_{1,1}| \, s \, t}\right), & \rho_{1,1} < 0, \\ I_0\left(2\sqrt{\rho_{1,1} \, s \, t}\right), & \rho_{1,1} \geq 0, \end{cases} \quad \text{uniformly in } (s,t) \in T_{1,1}, \tag{13}$$

*where $J_0$ and $I_0$ are the Bessel and modified Bessel functions of the first kind of order $0$, respectively.*

The truncation error decays as $\mathcal{O}((n!)^{-2})$ in the order $n$, making the series (13) highly effective for approximating $\kappa_{1,1}$, especially when $|\rho_{1,1}|$ is moderate (larger $|\rho_{1,1}|$ need higher truncation orders).

### 2.2.2 Recursive Neumann Series for Propagating the Signature Kernel Across All Tiles

The recursion (8) for the local power series coefficients $c_{k,l} \equiv (c_{k,l}^{(i,j)}) \in \ell_1(\mathbb{N}_0^2)$ on the remaining tiles starts from the base coefficients (12) and proceeds as follows.

For $k, l = 1, \ldots, \ell - 1$, define the (propagation) operators $\boldsymbol{T}_{k,l} : C(T_{k,l}) \to C(T_{k,l})$ by

$$(\boldsymbol{T}_{k,l} f)(s,t) = \int_{\tau_l}^t \int_{\sigma_k}^s \rho_{k,l} f(u,v) \, \mathrm{d}u \, \mathrm{d}v, \quad (s,t) \in T_{k,l} \tag{14}$$

(cf. Lemma 2.3), and set $T_{0,l} := \{0\} \times [\tau_l, \tau_{l+1}]$ and $T_{k,0} := [\sigma_k, \sigma_{k+1}] \times \{0\}$.

**Proposition 2.5.** *For each $k, l = 1, \ldots, \ell - 1$, the restricted kernel $\kappa_{k,l} = K|_{T_{k,l}}$ from* (9) *satisfies*

$$\kappa_{k,l} = \sum_{n=0}^{\infty} \boldsymbol{T}_{k,l}^n \left( \kappa_{k-1,l}^{(\sigma_k, \cdot)} + \kappa_{k,l-1}^{(\cdot, \tau_l)} - \kappa_{k-1,l-1}^{(\sigma_k, \tau_l)} \right) \quad \text{uniformly on } T_{k,l}, \tag{15}$$

*for $\kappa_{k,l}^{(\sigma,\tau)} \equiv \kappa_{k,l}(\sigma, \tau)$ and the 'curried' functions $\kappa_{k,l}^{(\sigma, \cdot)} : T_{k,l} \ni (u,v) \mapsto \kappa_{k,l}(\sigma, v)$ and $\kappa_{k,l}^{(\cdot, \tau)} : T_{k,l} \ni (u,v) \mapsto \kappa_{k,l}(u, \tau)$. Here, $\kappa_{0,l} := K|_{T_{0,l}}$ and $\kappa_{k,0} := K|_{T_{k,0}}$ and $\kappa_{0,0} \equiv K(0,0) = 1$.*

The identities in (15) yield the desired tilewise power-series representation (5)–(6). On each tile, the coefficients (7) are determined recursively from those on the tiles immediately to the left and below. The following example illustrates this.

---

[3]This is an assumption only for the moment – we will establish (10) as a provable identity in Lemma 2.4.

**Example 2.6** (Evaluating (15) on $T_{1,1}$, $T_{1,2}$, and $T_{2,1}$). *With $\sigma_i = \tau_i = \frac{i-1}{\ell-1}$ and $\kappa_{0,i} = \kappa_{i,0} = \kappa_{0,0} \equiv 1$, the recursion (15) gives on the bottom-left corner tile $T_{1,1}$ that*

$$\kappa_{1,1} = \sum_{n=0}^{\infty} \boldsymbol{T}_{1,1}^n 1 + \boldsymbol{T}_{1,1}^m 1 - \boldsymbol{T}_{1,1}^m 1 = \sum_{n=0}^{\infty} \boldsymbol{T}_{1,1}^n 1 \quad \text{uniformly on } T_{1,1}, \tag{16}$$

*where the operator $\boldsymbol{T}_{1,1} : f \longmapsto \left[ (s,t) \mapsto \int_0^t \int_0^s \rho_{1,1} f(u,v) \, \mathrm{d}u \, \mathrm{d}v \right]$ is applied repeatedly. Since $(\boldsymbol{T}_{1,1}^n 1)(s,t) = \frac{(\rho_{1,1} st)^n}{(n!)^2}$ for each $n \in \mathbb{N}_0$ (as can be readily verified by induction), the recursion (16) precisely recovers the expansion (13). On the adjacent tile $T_{1,2}$, the identity (15) yields*

$$\kappa_{1,2}(s,t) = \sum_{n=0}^{\infty} \left[ \boldsymbol{T}_{1,2}^n \kappa_{0,2}^{(\sigma_1,t)} + \boldsymbol{T}_{1,2}^n \kappa_{1,1}^{(s,\tau_2)} - \boldsymbol{T}_{1,2}^n \kappa_{0,1}^{(\sigma_1,\tau_2)} \right](s,t)$$

$$= \sum_{n=0}^{\infty} \left[ \boldsymbol{T}_{1,2}^n 1 + \sum_{i=0}^{\infty} \frac{\rho_{1,1}^i \tau_2^i}{(i!)^2} \boldsymbol{T}_{1,2}^n \tilde{s}^i - \boldsymbol{T}_{1,2}^n 1 \right](s,t) = \sum_{i,j=0}^{\infty} \frac{\rho_{1,1}^i \rho_{1,2}^j}{(i+j)! i! (\ell-1)^i} s^{i+j} (t-\tau_2)^j,$$

*where the last equality used the definition of $\tau_2$ and that, for each iteration index $n \in \mathbb{N}_0$,*

$$(\boldsymbol{T}_{1,2}^n \tilde{s}^i)(s,t) = \rho_{1,2} \int_{\tau_2}^t \int_0^s (\boldsymbol{T}^{n-1} \tilde{s}^i)(u,v) \, \mathrm{d}u \, \mathrm{d}v = \frac{\rho_{1,2}^n}{(i+1)^{\bar{n}} n!} s^{i+n} (t-\tau_2)^n \tag{17}$$

*with $(x)^{\bar{n}} := \prod_{i=0}^{n-1} (x+i)$, as one verifies immediately (Lemma A.2). Analogous computations show*

$$\kappa_{2,1}(s,t) = \sum_{k,l=0}^{\infty} \frac{\rho_{1,1}^l \rho_{2,1}^k}{(k+l)! l! (\ell-1)^l} (s-\sigma_2)^k t^{k+l}$$

*uniformly in $(s,t) \in T_{2,1}$.* ◇

The observation (17) is recorded as Lemma A.2 for later use. Given the recursion (15), we now need an explicit algorithm ($\phi$) to extract the coefficients in (8) and thus build the approximations in (5). The next section provides this (Propositions 2.7 and 2.8).

## 2.3 Computing the Neumann Series Coefficients

To algorithmically extract the power-series coefficients from the tilewise Neumann recursions (15), we encode the action of the propagation operators $\boldsymbol{T}_{k,l}$ on the monomial basis $\{s^i t^j \mid (i,j) \in \mathbb{N}_0^2\} \subset C(T_{k,l})$ via a simple Vandermonde scheme: Define the power map $\eta : [0,1] \to \ell_\infty(\mathbb{N}_0)$ by

$$\eta(r) = (r^i \mid i \in \mathbb{N}_0) \equiv (1, r, r^2, r^3, \cdots),$$

and, for $(c_{ij})_{i,j\geq 0} \in \ell_1(\mathbb{N}_0^2)$, define the doubly-infinite matrices $C \in \mathscr{L}(\ell_\infty(\mathbb{N}_0), \ell_1(\mathbb{N}_0))$ by

$$C \equiv (c_{i,j})_{i,j\geq 0} \; : \; a \equiv (a_j)_{j\geq 0} \longmapsto \left( \sum_{j\geq 0} c_{ij} a_j \right)_{i\geq 0} =: C \cdot a. \tag{18}$$

By the Weierstrass $M$-test, each such $C$ then induces a continuous function

$$C_{\langle \sigma, \tau \rangle} \; : \; [0,1]^2 \ni (s,t) \longmapsto \langle \eta(s), C\eta(t) \rangle = \sum_{i,j\geq 0} c_{ij} s^i t^j \in \mathbb{R}, \tag{19}$$

where $\langle \cdot, \cdot \rangle$ denotes the dual pairing between $\ell_\infty(\mathbb{N}_0)$ and $\ell_1(\mathbb{N}_0)$. The localised kernels (9) can then all be represented in the explicit form (19) for some recursively related coefficient matrices $C^{k,l}$:

**Proposition 2.7.** *For each $k, l \in \{1, \ldots, \ell-1\}$, there is $C^{k,l} \equiv (c_{k,l}^{i,j})_{i,j\geq 0} \in \ell_1(\mathbb{N}_0^2)$ such that*

$$\kappa_{k,l} = C_{\langle \sigma, \tau \rangle}^{k,l} \Big|_{T_{k,l}} \quad \text{and} \quad C^{k,l} = \sum_{n=0}^{\infty} C_n^{k,l} \quad \text{in } \ell_1(\mathbb{N}_0^2), \tag{20}$$

*with $\kappa_{k,l} = \lim_{m\to\infty} \left[ \sum_{n=0}^m C_n^{k,l} \right]_{\langle \sigma, \tau \rangle}$ uniformly on $T_{k,l}$. The sequence $(C_n^{k,l})_{n\geq 0} \subset \ell_1(\mathbb{N}_0^2)$ is*

$$\text{recursively defined by} \quad C_{n+1}^{k,l} = \rho_{k,l} L_{\sigma_k} C_n^{k,l} R_{\tau_l} \quad (n \in \mathbb{N}_0) \tag{21}$$

$$\text{with initial value} \quad C_0^{k,l} := \left( \alpha_i \delta_{i0} + \beta_j \delta_{0j} - \gamma \delta_{0i} \cdot \delta_{0j} \right)_{i,j\geq 0}, \tag{22}$$

*for* $(\alpha_i) := C^{k,l-1}\eta(\tau_l)$, $(\beta_i) := \left(C^{k-1,l}\right)^\dagger \eta(\sigma_k) \in \ell_1(\mathbb{N}_0)$ *and* $\gamma := \langle \eta(\sigma_k), C^{k-1,l}\eta(\tau_l)\rangle$ *and the boundary coefficients* $C^{0,\iota} = C^{\iota,0} := (\delta_{0i} \cdot \delta_{0j})_{i,j\geq 0}$ *for each* $\iota \in \mathbb{N}_0$. *Identities* (21) *use the matrices*

$$L_\sigma := (I - H(\sigma))\underline{S} \quad and \quad R_\tau := \underline{T}(I - G(\tau)),$$

$I := (\delta_{ij})_{i,j\geq 0}$, $H(\sigma) := (\sigma^j \delta_{i0})_{i,j\geq 0}$, $G(\tau) := (\tau^i \delta_{0j})_{i,j\geq 0}$, $\underline{S} :\overset{4}{=} \left(\frac{\delta_{i-1,j}}{i}\right)_{i,j\geq 0}$, $\underline{T} := \left(\frac{\delta_{i,j-1}}{j}\right)_{i,j\geq 0}$.

Numerically, the dominant cost in applying (15) is the double integration $\int_{\tau_l}^{\cdot\cdot} \int_{\sigma_k}^{\cdot\cdot}$, i.e., the multiplications $L_{\sigma_k}(\cdot)R_{\tau_l}$ in (21). This cost depends on the expansion center $(s_k, t_l)$ in the representation

$$\kappa_{k,l}(s,t) = \sum_{i,j=0}^\infty c_{k,l;(s_k,t_l)}^{i,j}(s - s_k)^i(t - t_l)^j \qquad ((s,t) \in T_{k,l}). \tag{23}$$

Proposition 2.7 establishes (23) for $(s_k, t_l) = (0,0)$ and yields a direct implementation via (20)–(21). The next result shows that centering instead at the tile corner $(s_k, t_l) = (\sigma_k, \tau_l)$ substantially reduces computation while preserving uniform convergence. In what follows, $\odot$ denotes the Hadamard product of doubly-infinite matrices (i.e., the entry-wise matrix multiplication $(a_{i,j})_{i,j\geq 0} \odot (b_{i,j})_{i,j\geq 0} := (a_{i,j}b_{i,j})_{i,j\geq 0}$), and we abbreviate $\ell' := \ell - 1$.

**Proposition 2.8.** *For each* $k, l \in \{1, \ldots, \ell'\}$, *the localised solution* (9) *has the tile-centered expansion*

$$\kappa_{k,l}(s,t) = \sum_{i,j=0}^\infty \tilde{c}_{i,j}^{(k,l)}(s - \sigma_k)^i(t - \tau_l)^j \quad for \quad \tilde{C}^{k,l} \equiv \left(\tilde{c}_{i,j}^{(k,l)}\right)_{i,j\geq 0} := A_{k,l} \odot B_{k,l} \odot W, \tag{24}$$

*uniformly in* $(s,t) \in T_{k,l}$, *where* $W \equiv (w_{i,j})_{i,j\geq 0}$ *with* $w_{i,j} := \frac{\left(\max(i,j) - \min(i,j)\right)!}{\max(i,j)! \min(i,j)!}$, *as well as* $A_{k,l} := \left(\rho_{k,l}^{\min(i,j)}\right)_{i,j\geq 0}$, *and the matrix* $B_{k,l} \equiv \left(b_{i,j}^{(k,l)}\right)_{i,j\geq 0}$ *is defined by*

$$b_{i,i+r}^{(k,l)} := \alpha_r^{(k,l)} \quad and \quad b_{i,i-r}^{(k,l)} := \beta_{|r|}^{(k,l)}, \quad for each \ (i,r) \in \mathbb{N}_0^2, \tag{25}$$

*with* $\beta_r^{(1,1)} = \alpha_r^{(1,1)} := \delta_{0,r}$ *for each* $r \in \mathbb{N}_0$, *and recursively* (*recalling* (18) *for notation*),

$$\left(\alpha_r^{(k,l)}\right)_{r\geq 0} := \tilde{C}^{k,(l-1)} \cdot \eta(1/\ell') \quad and \quad \left(\beta_r^{(k,l)}\right)_{r\geq 0} := \left[\tilde{C}^{(k-1),l}\right]^\dagger \cdot \eta(1/\ell'). \tag{26}$$

Equations (24), (25), and (26) define an efficient, tile-centered implementation of (8). Proposition A.3 provides rigorous a priori bounds for the induced Gram-matrix approximation error. The next section presents a numerical evaluation of this method—covering accuracy, memory usage, and runtime—and illustrates its applicability to downstream tasks on real data.

## 3 Numerical Experiments

We evaluate our method, `PowerSig`, in terms of accuracy, memory usage, and runtime. Specifically, we compute the self-signature kernel of randomly drawn two-dimensional Brownian motion paths on $[0,1]$ at increasing sampling frequencies, using sample lengths $\ell = 2^k + 1$ for $k \geq 0$, constrained only by GPU memory; comparisons are made against the state-of-the-art `KSig` library [19]. We use `KSig` both with its truncated signature kernel and with its PDE-based solver at the default dyadic order. Experiments were run on an NVIDIA RTX 4090 GPU (24 GB). Unless noted, `PowerSig` truncation order for the tile-center local series (24) is fixed at 7, although higher orders are equally feasible. Accuracy is reported as Mean Absolute Percentage Error (MAPE) relative to the `KSig` truncated signature kernel (order 1, truncation level 21), memory is peak GPU usage, and runtime is total execution time. Each point averages 10 independent runs.

**Accuracy.** Figure 2 compares the accuracy between `PowerSig` and the PDE-based solver from `KSig` across two-dimensional Brownian motion paths of length up to 513 (the maximum length manageable by the truncated signature kernel; left panel) and on two-dimensional fractional Brownian motion paths of fixed length 51 across decreasing Hurst indices (from 0.4 down to 0.005; right panel). `PowerSig`, despite employing only a modest truncation order, achieves superior accuracy and remarkably low error levels as length and irregularity ('roughness') of the input time series increase. `PowerSig`'s robust performance on time series with low Hurst indices further suggests significantly enhanced numerical stability of the method, particularly for highly irregular (rough) trajectories.

---

[4]Here and in the definition of $\underline{T}$ we adopt the convention $\frac{0}{0} := 0$.

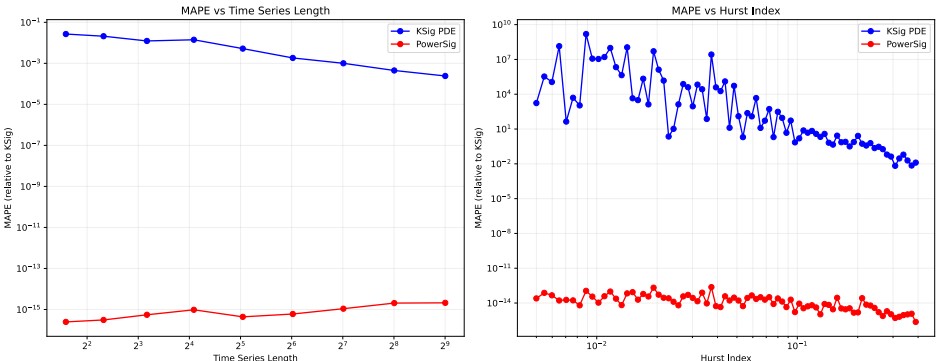

Figure 2: *Comparison of Mean Absolute Percentage Error (MAPE) between* `PowerSig` *and the PDE-based solver from* `KSig`*. Left: for two-dimensional Brownian motion paths on* $[0, 1]$ *across increasing path lengths* $\ell$*. Right: for two-dimensional fractional Brownian motions of fixed length* $\ell = 51$ *across increasingly irregular sample paths (decreasing Hurst index, swept through progressively rougher regimes); the right panel reports MAPE relative to the signature kernel truncated at level 180.*

**Memory Usage and Runtime.**    Figure 3 highlights the practical advantages of `PowerSig` in terms of GPU memory usage and runtime. Specifically, its localized, tile-based computation drastically reduces memory overhead, enabling computations on paths of length $\ell = 524\,289$ with under $720\,\text{MB}$ GPU memory, which is orders of magnitude lower than both PDE- and dynamic-programming-based methods. The inherent sparsity of the propagated Neumann-series expansions enables efficient memory management, allowing `PowerSig` to handle substantially larger-scale problems and overcome the storage bottleneck associated with full-grid methods.

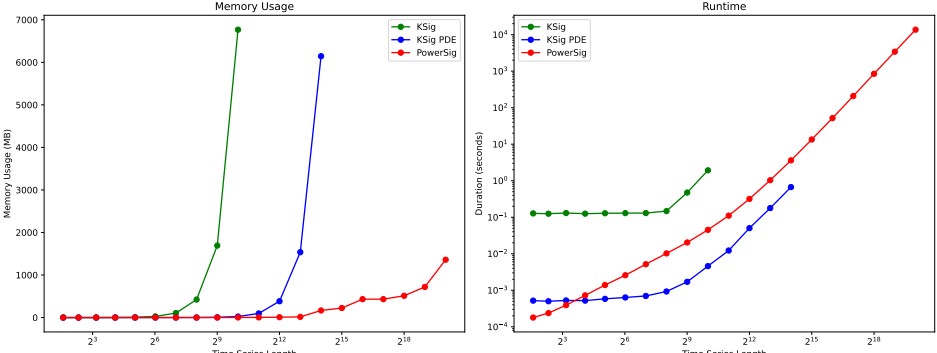

Figure 3: *Peak GPU memory usage (left) and runtime (right) for computing the signature kernel on two-dimensional Brownian motion paths, comparing* `PowerSig` *with the truncated-signature (KSig) and PDE-based (KSig PDE) solvers.* `PowerSig` *achieves substantially lower memory consumption and maintains computational feasibility for large* $\ell$*, well beyond the limits of the alternative methods.*

**Time Complexity**    `PowerSig` retains the $\mathcal{O}(\ell^2 d)$ runtime scaling of existing methods but significantly improves space complexity to $\mathcal{O}(\ell P)$, where $P$ (polynomial truncation order) is typically much smaller than $\ell$. By storing only on a single diagonal of coefficient blocks, `PowerSig` enables the processing of much longer paths than previously feasible with existing approaches.

**Empirical Evaluation on Real and Large-Scale Settings**

Beyond the above benchmarks, we also assess downstream performance and compare `PowerSig` with recent low-rank and random Fourier Feature (RFF) approximations. Unless noted otherwise, we use the default truncation policy and report averages over multiple independent runs. All figures,

implementation details, and full hyperparameter grids appear in the supplement (downstream figures have been moved to Section B.1 of the supplement to comply with page limits).

(A) *Bitcoin price regression (Salvi et al. [17]).* Figure 4 shows train and test fits (two-day rolling average) for kernel-ridge regression (MAPE) on the public bitcoin pricing dataset featured in Salvi et al. in [17]. On the test set, `PowerSig` attains $2.81\%$ MAPE versus $3.23\%$ for the (RBF-assisted) `KSig-PDE`. For the underlying Gram-matrix construction, peak memory for `KSig-PDE` scales as $\mathcal{O}(N^2\ell^2)$ (with $N$ windows and window length $\ell$), whereas `PowerSig` uses only $\mathcal{O}(\ell^2)$. For the present setup ($N = 299$, $\ell = 36$), this extrapolates to roughly $\sim 1.6$,GB for `KSig-PDE` versus $\sim !0.038$,MB for `PowerSig`. This illustrates that the near-exact regime enabled by `PowerSig`'s linear-in-length memory profile yields tangible predictive gains at far lower resource cost, with particularly clear benefits even at short window lengths.

(B) *UEA Eigenworms classification and RFF/low-rank baselines.* We benchmark `PowerSig` and `KSig-PDE` against linear/RBF kernel SVMs and the recent specialized RFF-based method `RFSF-TRP` from [20] on the standard Eigenworms dataset with input window lengths $L \in \{16, 32, \ldots, 1024\}$. As shown in Figure 5, `PowerSig` (and `KSig-PDE` up to $L = 128$ before OOM) remains competitive and rises to $61.1\%$ accuracy at $L = 1024$, whereas `RFSF-TRP` attains a slightly higher peak of $62.5\%$ at $L = 128$ but exhausts memory for larger $L$, consistent with the storage advantages in Figure 3. These results show that substantially extending the input window—enabled here at scale by `PowerSig`—can narrow performance gaps often ascribed to inductive bias while maintaining feasibility.

(C) *Long-horizon periodic signals (industrial/sensing proxy).* Motivated by predictive maintenance (near-periodic gearbox/turbine vibrations) and narrow-band I/Q radio signals, we generate synthetic near-periodic time series with adjustable period length. For representative instances (see, e.g., Figure 7), SVM-regression error decreases monotonically as input windows span multiple periods. As shown in Figure 6, `PowerSig` sustains this behavior at window lengths beyond the reach of conventional or low-rank signature-kernel methods, while peak memory grows linearly with window length.

(D) *High-dimensional scaling.* Complementing Figure 3, we fix path length $\ell = 4096$ and vary dimension $d$ from 2 to 8192. Figure 8 shows stable accuracy and near-perfect linear runtime from $d = 64$ to $4096$ (slightly sublinear outside), with memory following our one-strip tiling profile. This corroborates the practicality of `PowerSig` in high-dimensional sensing and multivariate finance.

Overall, across real regression and classification tasks and stress tests in length and dimension, `PowerSig` delivers competitive or superior accuracy, strong robustness to rough and long inputs, and demonstrated scalability, while using substantially less memory than alternative methods.

# 4 Conclusion

We introduced `PowerSig`, a method for computing signature kernels of piecewise-linear time series via localized Neumann–series expansions. Recasting the kernel-defining Goursat PDE as a Volterra equation yields uniformly convergent, tile-centered power-series expansions of the kernel that propagate only boundary data along a directed tile graph and admit efficient adaptive per-tile truncation (Lemma 2.3, Prop. 2.5, Prop. 2.8, Prop. A.3). The resulting design achieves linear-in-length memory $\mathcal{O}(\ell P)$, preserves the standard $\mathcal{O}(\ell^2 d)$ runtime, and supports straightforward parallelism.

Empirically, `PowerSig` matches or exceeds state-of-the-art PDE- and DP-based solvers in accuracy, remains stable on highly irregular (low-Hurst) inputs, and scales to path lengths previously infeasible on commodity GPUs. On downstream tasks it delivers competitive or improved predictive performance at substantially lower memory cost.

Future work includes tighter extraction of tile-boundary coefficients, adaptive scheduling across tiles to leverage hardware concurrency, and refined truncation policies guided by local roughness. Additional directions include extending beyond piecewise-linear interpolation (e.g., to higher-order segments or learned segment maps), integrating certified a posteriori error control, and broadening applications in finance, sensing, and long-horizon sequence modeling.

## Acknowledgements

The authors thank Csaba Tóth for helpful discussions on benchmarking signature kernel downstream tasks. The authors are also grateful to the program chair and three anonymous reviewers for their helpful and constructive comments and suggestions. A.S. acknowledges funding from the Bavarian State Ministry of Sciences and the Arts in the framework of the bidt Graduate Center for Postdocs.

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

# A  Mathematical Proofs

## A.1  Proof of Lemma 2.3

*Proof of Lemma 2.3.* By definition of $r(\boldsymbol{T}_\varrho)$, we need to show that the spectrum of $\boldsymbol{T}_\varrho$ is $\sigma(\boldsymbol{T}_\varrho) = \{0\}$, i.e., that zero is the only element in the spectrum of $\boldsymbol{T}_\varrho$. This is equivalent to establishing that for all $\lambda \in \mathbb{C}\backslash\{0\}$, the operator $\lambda\,\mathrm{id} - \boldsymbol{T}_\varrho$ is a bijection with a bounded inverse. Here, id denotes the identity on $C([0,1]^2)$. To show this, we note that it suffices to prove that

$$\lambda\,\mathrm{id} - \boldsymbol{T}_\varrho : C(I) \to C(I)$$

is a bijection: since $C(I)$ is a Banach space, the inverse mapping theorem implies that in this case, the inverse is a bounded operator. Thus, it is left to show that

$$(\lambda\,\mathrm{id} - \boldsymbol{T}_\varrho)\,K = F \tag{27}$$

is uniquely solvable in $C(I)$ for all $\lambda \in \mathbb{C}\backslash\{0\}$ and all $F \in C(I)$.

One way to establish this result is via the Picard iteration, through proving that (see [15]) the Picard iterates $K_n$ given recursively by

$$K_{n+1}(s,t) = F(s,t) + \int_0^s \int_0^t \varrho(u,v) K_n(u,v)\,\mathrm{d}u\,\mathrm{d}v, \quad K_0(s,t) \equiv 0,$$

form a Cauchy sequence in $(C(I), \|\cdot\|_\infty)$:

$$|K_{n+1}(s,t) - K_n(s,t)| \le \left(\frac{1}{2}\right)^n \|F\|_\infty\, e^{\beta(s+t)}, \quad (s,t) \in I,$$

where $\beta = (2\,\|\varrho\|_\infty)^{1/2}$.

This guarantees the existence of the limit $K_* := \lim_{n\to\infty} K_n$ as an element in $C(I)$. Since each $K_n$ is bounded by

$$|K_n(s,t)| = |K_n(s,t) - K_0(s,t)| \le \sum_{i=0}^{n-1} |K_{i+1}(s,t) - K_i(s,t)|,$$

$$\le \sum_{i=0}^{n-1} 2^{-i} \|F\|_\infty\, e^{\beta(s+t)} \le 2\,\|F\|_\infty\, e^{\beta(s+t)},$$

which is an integrable function on $I$, it follows by dominated convergence that

$$K_*(s,t) = \lim_{n\to\infty} K_{n+1}(s,t) = F(s,t) + \int_0^s \int_0^t \lim_{n\to\infty} \varrho(u,v) K_n(u,v)\,\mathrm{d}u\,\mathrm{d}v,$$

$$= F(s,t) + \int_0^s \int_0^t \varrho(u,v) K_*(u,v)\,\mathrm{d}u\,\mathrm{d}v,$$

so that $K_*$ solves the integral equation (27). As noted in [15], uniqueness can be established with another proof by induction. We give a precise argument for the convenience of the reader.

For this, suppose that there exists another solution $K \in C(I)$ to (27) different from $K_*$. Then,

$$K_{n+1}(s,t) - K(s,t) = \int_0^s \int_0^t \varrho(u,v)\,(K_n(u,v) - K(u,v))\,\mathrm{d}u\,\mathrm{d}v. \tag{28}$$

As suggested in [15], showing

$$|K_n(s,t) - K(s,t)| \le 2^{-n} \, \|K\|_\infty \, e^{\beta(s+t)},$$

is sufficient for establishing uniqueness of the solution of (27). For the proof by induction, note that the case $n = 0$ holds trivially since $K_0 \equiv 0$. The induction step $n \mapsto n+1$ is also straightforward:

$$
\begin{aligned}
|K_{n+1}(s,t) - K(s,t)| &= \left| \int_0^s \int_0^t \varrho(u,v) \left( K_n(u,v) - K(u,v) \right) \mathrm{d}u \, \mathrm{d}v \right|, \\
&\le \|\varrho\|_\infty \, 2^{-n} \, \|K\|_\infty \int_0^s \int_0^t e^{\beta(u+v)} \, \mathrm{d}u \, \mathrm{d}v, \\
&\le \|\varrho\|_\infty \, 2^{-n} \, \|K\|_\infty \, \beta^{-2} \left( e^{\beta s} - 1 \right) \left( e^{\beta t} - 1 \right), \\
&\le 2^{-(n+1)} \, \|K\|_\infty \, e^{\beta(s+t)},
\end{aligned}
$$

where the first line is an application of (28). Therefore, in $C(I)$, a solution to (27) always exists and is unique. Together with employing the inverse mapping theorem, this concludes that the spectral radius of $\boldsymbol{T}_\varrho$ is zero. $\qquad\square$

## A.2 Proof of Proposition 2.2

*Proof.* We can reformulate (2) as the equivalent integral equation (4), which in turn is equivalent to

$$(\mathrm{id} - \boldsymbol{T}_{\rho_{\boldsymbol{x},\boldsymbol{y}}})K = u_0$$

for id the identity on $C([0,1]^2)$ and $u_0$ the constant one-function $u_0 \equiv 1$ and the integral operator

$$\boldsymbol{T}_{\rho_{\boldsymbol{x},\boldsymbol{y}}} : \left( C([0,1]^2), \|\cdot\|_\infty \right) \longrightarrow \left( C([0,1]^2), \|\cdot\|_\infty \right), \; f \longmapsto \left[ (s,t) \mapsto \int_0^t \int_0^s \rho_{\boldsymbol{x},\boldsymbol{y}}(u,v) \, f(u,v) \, \mathrm{d}u \, \mathrm{d}v \right].$$

The operator $\boldsymbol{T}_{\rho_{\boldsymbol{x},\boldsymbol{y}}}$ is clearly linear and bounded, and has spectral radius zero, $r(\boldsymbol{T}_{\rho_{\boldsymbol{x},\boldsymbol{y}}}) = 0$, by Lemma 2.3. Consequently (cf. [16, Theorem VI.6]), we have $\lim_{n \to \infty} \left\| \boldsymbol{T}_{\rho_{\boldsymbol{x},\boldsymbol{y}}}^n \right\|^{1/n} = r(\boldsymbol{T}_{\rho_{\boldsymbol{x},\boldsymbol{y}}}) = 0$, implying that

$$\forall q \in (0,1) : \exists n_q \in \mathbb{N} : \|\boldsymbol{T}_{\rho_{\boldsymbol{x},\boldsymbol{y}}}^n\| < q^n, \; \forall n \ge n_q.$$

Consequently, the Neumann series $\sum_{n=0}^\infty \boldsymbol{T}_{\rho_{\boldsymbol{x},\boldsymbol{y}}}^n$ is $\|\cdot\|$-convergent and, hence,

$$(\mathrm{id} - \boldsymbol{T}_{\rho_{\boldsymbol{x},\boldsymbol{y}}}) \text{ is invertible} \quad \text{with} \quad (\mathrm{id} - \boldsymbol{T}_{\rho_{\boldsymbol{x},\boldsymbol{y}}})^{-1} = \sum_{n=0}^\infty \boldsymbol{T}_{\rho_{\boldsymbol{x},\boldsymbol{y}}}^n.$$

This, however, implies that $K$ is indeed the only solution of (4), additionally satisfying

$$K = (\mathrm{id} - \boldsymbol{T}_{\rho_{\boldsymbol{x},\boldsymbol{y}}})^{-1} u_0 = \sum_{n=0}^\infty \boldsymbol{T}_{\rho_{\boldsymbol{x},\boldsymbol{y}}}^n u_0.$$

$\qquad\square$

*Remark* A.1. Note that the $n$-th Picard iterate is related to the Neumann series via

$$K_n = \sum_{i=0}^n \boldsymbol{T}_{\rho_{\boldsymbol{x},\boldsymbol{y}}}^i K_1.$$

We remark that this convergence result is independent of any bound on $\rho_{\boldsymbol{x},\boldsymbol{y}}$ and extends a classical result for Volterra equations in one dimension [7, Theorem 4.1]. In particular, $\boldsymbol{T}_{\rho_{\boldsymbol{x},\boldsymbol{y}}}$ need not be a contractive mapping. Repetition of the above argument similarly results in the converegence of $\sum_{n=0}^\infty k^{-n-1} \boldsymbol{T}_{\rho_{\boldsymbol{x},\boldsymbol{y}}}^n$, which corresponds to the fixed point iteration for solving $(k \, \mathrm{id} - \boldsymbol{T}_{\rho_{\boldsymbol{x},\boldsymbol{y}}})K = f$, for any $k \ne 0$ and any $f \in C(I)$. $\qquad\diamond$

### A.3 Proof of Lemma 2.4

*Proof.* The (bivariate) power series $k_{1,1} : (s,t) \mapsto \sum_{i=0}^{\infty} \frac{\rho_{1,1}^i}{(i!)^2} s^i t^i$ converges uniformly absolutely on $[0,1]^2 \supset T_{1,1}$, since $\sum_{i=0}^{\infty} \left| \frac{\rho_{1,1}^i}{(i!)^2} s^i t^i \right| \leq e^{|\rho_{1,1}|}$ for all $(s,t) \in [0,1]^2$. In particular, $k_{1,1}$ is partially differentiable with mixed derivatives $(\partial_s \partial_t k_{1,1})(s,t) = \rho_{1,1} \sum_{i=1}^{\infty} \frac{\rho_{1,1}^{i-1} i^2}{(i!)^2} (st)^{i-1} = \rho_{\boldsymbol{x},\boldsymbol{y}}(s,t) k_{1,1}(s,t)$, for all interior points $(s,t)$ of $T_{1,1}$. Thus, $k_{1,1}$ solves the boundary value problem (2) on the tile $\mathcal{D}_{1,1}$, as does $K$. By the uniqueness of solutions to (2), we conclude that $k_{1,1}|_{T_{1,1}} = K|_{T_{1,1}}$, which establishes (13). $\qquad\square$

### A.4 Proof of Proposition 2.5

*Proof.* Let us note first that any point $(s,t) \in [0,1]^2$ can be decomposed as

$$(s,t) = (\tilde{s},\tilde{t}) + (\sigma_{k(s)}, \tau_{l(t)})$$

with $(\tilde{s},\tilde{t}) \in \hat{T}_{1,1}$ and position indices $(k(s),l(t)) := (\lfloor s(\ell-1) \rfloor + 1, \lfloor t(\ell-1) \rfloor + 1) \in \{1,\ldots,\ell\}^2$ determined by the location of $(s,t)$ within the tiling $(\hat{T}_{k,l})$. Hence, and by the Volterra identity (4),

$$K(s,t) = K(\sigma_{k(s)}, t) + K(s, \tau_{l(t)}) - K(\sigma_{k(s)}, \tau_{l(t)}) + \int_{\tau_{l(t)}}^{t} \int_{\sigma_{k(s)}}^{s} \rho_{k(s),l(t)} K(u,v)\, \mathrm{d}u\, \mathrm{d}v \quad (29)$$

for all $(s,t) \in [0,1]^2$. Defining the 'boundary maps' $\gamma_{k,l} : T_{k,l} \to \mathbb{R}$ $(k,l = 1,\ldots,\ell-1)$ by

$$\gamma_{k,l}(s,t) = K(\sigma_k, t) + K(s, \tau_l) - K(\sigma_k, \tau_l), \quad (s,t) \in T_{k,l}, \quad (30)$$

the identity (29) can be expressed as

$$K(s,t) = \gamma_{k(s),l(t)}(s,t) + (\boldsymbol{T}_{k(s),l(t)} \kappa_{k(s),l(t)})(s,t), \quad (s,t) \in [0,1]^2. \quad (31)$$

Since for each $(s,t) \in \hat{T}_{k,l}$, it holds that $(k(s),l(t)) = (k,l)$, equation (31) further is equivalent to the following $(k,l)$-indexed system of identities in $C(T_{k,l})$,

$$(\mathrm{id} - \boldsymbol{T}_{k,l}) \kappa_{k,l} = \gamma_{k,l} \quad (k,l = 1,\ldots,\ell-1). \quad (32)$$

From Lemma 2.3 and basic operator theory (cf. [18, Thm. 2.9]), we know the identities (32) are invertible for $\kappa_{k,l}$ and the respective inverse operators can be written as a Neumann series in $T_{k,l}$,

$$\kappa_{k,l} = (\mathrm{id} - \boldsymbol{T}_{k,l})^{-1} \gamma_{k,l} = \sum_{n=0}^{\infty} \boldsymbol{T}_{k,l}^n \gamma_{k,l}, \quad (33)$$

where the above series converges wrt. $\|\cdot\|_{\infty;T_{k,l}}$, the sup-norm on $C(T_{k,l})$.

For the recursive structure of the $\kappa_{k,l}$-identities (33), note that, for any fixed $(s,t) \in T_{k,l}$, we have $(\sigma_k, t) \in T_{k-1,l} \cap T_{k,l}$ and $(s, \tau_l) \in T_{k,l} \cap T_{k,l-1}$, where by definition $T_{0,l} = \{0\} \times [\tau_l, \tau_{l+1}]$ and $T_{k,0} = [\sigma_l, \sigma_{l+1}] \times \{0\}$ and $T_{0,0} := \{(0,0)\}$. Consequently, the boundary map (30) evaluates to

$$\gamma_{k,l}(s,t) = \kappa_{k-1,l}(\sigma_k, t) + \kappa_{k,l-1}(s, \tau_l) - \kappa_{k-1,l-1}(\sigma_k, \tau_l), \quad (s,t) \in T_{k,l}, \quad (34)$$

with $\kappa_{0,l} := K|_{T_{0,l}}$, $\kappa_{k,0} := K|_{T_{k,0}}$, $\kappa_{0,0} \equiv 1$. Combining (34) and (33) proves (15). $\qquad\square$

**Lemma A.2.** *Let $\varphi_\sigma^{(l)} : [0,1]^2 \ni (s,t) \mapsto (s-\sigma)^l$ and $\varphi_\tau^{(l)} : [0,1]^2 \ni (s,t) \mapsto (t-\tau)^l$, for any given $l \in \mathbb{N}_0$. Then for each $n \in \mathbb{N}_0$ and all $\mu,\nu \in \{1,\ldots,\ell-1\}$, we have that*

$$\boldsymbol{T}_{\mu,\nu}^n \big( \varphi_{\sigma_\mu}^{(l)} \big|_{T_{\mu,\nu}} \big)(s,t) = \frac{\rho_{\mu,\nu}^n}{(l+1)^{\bar{n}} n!} (s-\sigma_\mu)^{l+n} (t-\tau_\nu)^n, \quad (s,t) \in T_{\mu,\nu}, \quad \text{and}$$

$$\boldsymbol{T}_{\mu,\nu}^n \big( \varphi_{\tau_\nu}^{(l)} \big|_{T_{\mu,\nu}} \big)(s,t) = \frac{\rho_{\mu,\nu}^n}{(l+1)^{\bar{n}} n!} (s-\sigma_\mu)^n (t-\tau_\nu)^{l+n}, \quad (s,t) \in T_{\mu,\nu}.$$

*Proof of Lemma A.2.* This follows immediately by induction. Indeed: the case $n = 0$ is clear, and for $n \in \mathbb{N}$ we get

$$
\begin{aligned}
\boldsymbol{T}_{\mu,\nu}^n\big(\varphi_{\sigma_\mu}^{(l)}\big|_{T_{\mu,\nu}}\big)(s,t) &= \rho_{\mu,\nu} \int_{\tau_\nu}^t \int_{\sigma_\mu}^s \boldsymbol{T}_{\mu,\nu}^{n-1}\big(\varphi_{\sigma_\mu}^{(l)}\big|_{T_{\mu,\nu}}\big)(u,v)\, \mathrm{d}u\, \mathrm{d}v \\
&= \rho_{\mu,\nu} \int_{\tau_\mu}^t \int_{\sigma_\nu}^s \frac{\rho_{\mu,\nu}^{n-1}}{(l+1)^{\overline{n-1}}(n-1)!}(u - \sigma_\mu)^{l+n-1}(v - \tau_\nu)^{n-1}\, \mathrm{d}u\, \mathrm{d}v \\
&= \frac{\rho_{\mu,\nu}^n}{(l+1)^{\overline{n-1}}(n-1)!}\int_{\tau_\nu}^t (v - \tau_\nu)^{n-1}\, \mathrm{d}v \int_{\sigma_\mu}^s (u - \sigma_\mu)^{l+n-1}\, \mathrm{d}u \\
&= \frac{\rho_{\mu,\nu}^n}{(l+1)^{\overline{n-1}}(n-1)!}\frac{(t - \tau_\nu)^n}{n}\frac{(s - \sigma_\mu)^{l+n}}{l+n} \quad \text{for each } (s,t) \in T_{\mu,\nu},
\end{aligned}
$$

as claimed. The proof for $\boldsymbol{T}_{\mu,\nu}^n\big(\varphi_{\tau_\nu}^{(l)}\big|_{T_{k,l}}\big)$ is entirely analogous. $\qquad\square$

### A.5   Proof of Proposition 2.7

*Proof.* We proceed by induction on the tile position $(k,l)$. For this, note first that, for all $\sigma, \tau \geq 0$,

$$
\begin{aligned}
L_\sigma &= \big(\delta_{ij} - \sigma^j \delta_{i0}\big)_{i,j\geq 0} \cdot \Big(\tfrac{\delta_{i-1,j}}{i}\Big)_{i,j\geq 0} = \Big(\textstyle\sum_{k=0}^\infty \big(\tfrac{\delta_{ik} - \sigma^k \delta_{i0}}{k}\big)\delta_{k-1,j}\Big)_{i,j\geq 0} \\
&= \Big(\tfrac{\delta_{i,j+1} - \sigma^{j+1}\delta_{i0}}{j+1}\Big)_{i,j\geq 0} =: \big(\ell_{ij}(\sigma)\big)_{i,j\geq 0}, \quad \text{and} \tag{35}\\
R_\tau &= \Big(\tfrac{\delta_{i,j-1}}{j}\Big)_{i,j\geq 0} \cdot \big(\delta_{ij} - \tau^i \delta_{0j}\big)_{i,j\geq 0} = \Big(\textstyle\sum_{k=0}^\infty \big(\tfrac{\delta_{kj} - \tau^k \delta_{0j}}{k}\big)\delta_{i,k-1}\Big)_{i,j\geq 0} \\
&= \Big(\tfrac{\delta_{i,j-1} - \tau^{i+1}\delta_{0j}}{i+1}\Big)_{i,j\geq 0} =: \big(r_{ij}(\tau)\big)_{i,j\geq 0}. \tag{36}
\end{aligned}
$$

Then for the base case $(k,l) = (1,1)$, we have that $\big(C_{\langle\sigma,\tau\rangle}^{0,l} = C_{\langle\sigma,\tau\rangle}^{k,0} = C_{\langle\sigma,\tau\rangle}^{0,0} \equiv 1$ and hence) $C_0^{1,1} = (\delta_{0i} \cdot \delta_{0j})_{i,j\geq 0}$ and further, by (21), that

$$
\begin{aligned}
L_0 C_0^{1,1} R_0 &= (\ell_{ij}(0))_{i,j\geq 0} \cdot \big(\textstyle\sum_{n=0}^\infty (\delta_{0i} \cdot \delta_{0n})r_{nj}(0)\big)_{i,j\geq 0} = (\ell_{ij}(0))_{i,j\geq 0} \cdot \big(\delta_{0i} r_{0j}(0)\big)_{i,j\geq 0} \\
&= \big(\textstyle\sum_{n=0}^\infty \ell_{in}(0)r_{0j}(0) \cdot \delta_{0n}\big)_{i,j\geq 0} = \big(\ell_{i0}(0)r_{0j}(0)\big)_{i,j\geq 0} = \big(\delta_{i,1} \cdot \delta_{1,j}\big)_{i,j\geq 0}.
\end{aligned}
$$

Hence $\big[C_1^{1,1}\big]_{\langle\sigma,\tau\rangle} \overset{(21)}{=} \big[\rho_{1,1}L_0 C_0^{1,1} R_0\big]_{\langle\sigma,\tau\rangle} = \rho_{1,1}st$, whence $(20)\big|_{(k,l)=(1,1)}$ holds by Lemma 2.4 (and via induction on $n$) if, for any fixed $n \in \mathbb{N}_{\geq 2}$,

$$
\Big(\tfrac{\rho_{1,1}^n}{(n!)^2}\delta_{in}\delta_{nj}\Big)_{i,j\geq 0} = \rho_{1,1}L_0\Big(\tfrac{\rho_{1,1}^{n-1}}{((n-1)!)^2}\delta_{i,n-1}\delta_{n-1,j}\Big)_{i,j\geq 0}R_0. \tag{37}
$$

Denoting $\alpha_{ij} := \tfrac{\rho_{1,1}^{n-1}}{((n-1)!)^2}\delta_{i,n-1}\delta_{n-1,j}$ for each $i, j \geq 0$, the right-hand side of (37) reads

$$
\rho_{1,1}L_0 \cdot \big(\textstyle\sum_{\mu=0}^\infty \alpha_{i\mu}r_{\mu j}(0)\big)_{i,j\geq 0} = \rho_{1,1}\big(\textstyle\sum_{\nu,\mu=0}^\infty \ell_{i\nu}(0)\alpha_{\nu\mu}r_{\mu j}(0)\big)_{i,j\geq 0}. \tag{38}
$$

Now by (35) and (36), we obtain for any fixed $i, j, \nu, \mu \geq 0$ that

$$
\ell_{i\nu}(0)\alpha_{\nu\mu}r_{\mu j}(0) = \alpha_{\nu\mu} \cdot \tfrac{\delta_{i,\nu+1}}{\nu+1}\tfrac{\delta_{\mu,j-1}}{\mu+1} = \tfrac{\alpha_{i-1,j-1}}{ij} \cdot \delta_{i-1,\nu}\delta_{\mu,j-1}.
$$

This allows us to evaluate (38) to

$$
\rho_{1,1}\big(\textstyle\sum_{\nu,\mu=0}^\infty \ell_{i\nu}(0)\alpha_{\nu\mu}r_{\mu j}(0)\big)_{i,j\geq 0} = \rho_{1,1}\big(\alpha_{i-1,j-1}\big)_{i,j\geq 0} = \rho_{1,1}\Big(\tfrac{\rho_{1,1}^{n-1}}{((n-1)!)^2 ij}\delta_{in}\delta_{nj}\Big)_{i,j\geq 0},
$$

proving (37) as desired.

With the base case $(k,l) = (1,1)$ thus established, take now any $(k,l) \in \{1, \ldots, \ell - 1\}^2$ with $k + l > 2$, and suppose that (20) holds for $(k-1,l)$ and $(k,l-1)$ and $(k-1,l-1)$ (induction hypothesis). Then $\kappa_{\tilde{k},\tilde{l}} = C_{\langle\sigma,\tau\rangle}^{\tilde{k},\tilde{l}}\big|_{T_{\tilde{k},\tilde{l}}}$ for each $(\tilde{k},\tilde{l}) \in \{(k-1,l),(k,l-1),(k-1,l-1)\}$, whence and by Proposition 2.5 we have that

$$
\kappa_{k,l} = \sum_{n=0}^\infty \boldsymbol{T}_{k,l}^n\big(\langle \eta(\sigma_k), C^{k-1,l}\eta(\cdot)\rangle + \langle \eta(\cdot), C^{k,l-1}\eta(\tau_l)\rangle + \langle \eta(\sigma_k), C^{k-1,l-1}\eta(\tau_l)\rangle\big) \tag{39}
$$

uniformly on $T_{k,l}$. Since $\boldsymbol{T}_{k,l}^0 = \mathrm{id}|_{C(T_{k,l})}$, the $0^{\mathrm{th}}$ summand in (39) reads

$$\langle \eta(\sigma_k), C^{k-1,l}\eta(\cdot)\rangle + \langle \eta(\cdot), C^{k,l-1}\eta(\tau_l)\rangle + \langle \eta(\sigma_k), C^{k-1,l-1}\eta(\tau_l)\rangle = \left[C_0^{k,l}\right]_{\langle\sigma,\tau\rangle}$$

for the initial matrix $C_0^{k,l}$ from (22). Consequently, the claim (20) follows if, for any fixed $n \in \mathbb{N}$,

$$\left[C_{n+1}^{k,l}\right]_{\langle\sigma,\tau\rangle}\bigg|_{T_{k,l}} = \boldsymbol{T}_{k,l}\left(\left[C_n^{k,l}\right]_{\langle\sigma,\tau\rangle}\bigg|_{T_{k,l}}\right). \tag{40}$$

Abbreviating $u_{n+1} := \boldsymbol{T}_{k,l}\left(\left[C_n^{k,l}\right]_{\langle\sigma,\tau\rangle}\bigg|_{T_{k,l}}\right)$ and $C_n^{k,l} =: (c_{ij})_{i,j\geq 0}$, note that by definition,

$$u_{n+1}(s,t) = \rho_{k,l}\sum_{i,j\geq 0}c_{ij}\int_{\tau_l}^t\int_{\sigma_k}^s \tilde{s}^i\tilde{t}^j\,\mathrm{d}\tilde{s}\,\mathrm{d}\tilde{t} = \rho_{k,l}\sum_{i,j\geq 0}\frac{c_{ij}}{(i+1)(j+1)}(s^{i+1}-\sigma_k^{i+1})(t^{j+1}-\tau_l^{j+1}) \tag{41}$$

for each $(s,t) \in T_{k,l}$ (see (14) and (19)). Abbreviating $(\hat{c}_{ij}) := \left(\frac{c_{ij}}{(i+1)(j+1)}\right)_{i,j\geq 0}$ and using that

$$(s^{i+1}-\sigma_k^{i+1})(t^{j+1}-\tau_l^{j+1}) = s^{i+1}t^{j+1} - \sigma_k^{i+1}t^{j+1} - \tau_l^{j+1}s^{i+1} + \sigma_k^{i+1}\tau_l^{j+1}$$

for all $i,j \geq 0$, we can immediately rewrite (41) as $u_{k+1} = \tilde{C}_{\langle\sigma,\tau\rangle}\bigg|_{T_{k,l}}$ for the coefficient matrix

$$\tilde{C} := \rho_{k,l}\left(\hat{c}_{i-1,j-1} - \hat{\gamma}_{i-1|:}\cdot\delta_{0j} - \hat{\gamma}_{:|j-1}\cdot\delta_{i0} + \hat{\gamma}_{:|:}\cdot\delta_{i0}\delta_{0j}\right)_{i,j\geq 0},$$

where $\hat{\gamma}_{i|:} := \sum_{j=0}^\infty \hat{c}_{ij}\tau_l^{j+1}$ and $\hat{\gamma}_{:|j} := \sum_{i=0}^\infty \hat{c}_{ij}\sigma_k^{i+1}$ $(i,j\geq 1)$ and $\hat{\gamma}_{-1|:} = \hat{\gamma}_{:|-1} = \hat{c}_{-1,j} = \hat{c}_{i,-1} := 0$ $(i,j\geq 0)$, and $\hat{\gamma}_{:|:} := \sum_{i,j\geq 0}\sigma_k^{i+1}\tau_l^{j+1}$. Hence, the desired identity (40) follows if

$$\tilde{C} = \rho_{k,l}L_{\sigma_k}C_n^{k,l}R_{\tau_l}. \tag{42}$$

For a proof of (42), note simply that, by definition of $L_\sigma$ and $R_\tau$ (cf. (35) and (36)),

$$L_{\sigma_k}C_n^{k,l}R_{\tau_l} = \left(\sum_{\mu,\nu\geq 0}\ell_{i\mu}(\sigma_k)c_{\mu\nu}r_{\nu j}(\tau_l)\right)_{i,j\geq 0}$$

$$= \left(\sum_{\mu,\nu\geq 0}(\delta_{i,\mu+1} - \sigma_k^{\mu+1}\delta_{i0})\frac{c_{\mu\nu}}{(\mu+1)(\nu+1)}(\delta_{\nu,j-1} - \tau_l^{\nu+1}\delta_{0j})\right)_{i,j\geq 0}$$

$$= \left(\hat{c}_{i-1,j-1} - \sum_{\nu=0}^\infty\hat{c}_{i-1,\nu}\tau_l^{\nu+1}\delta_{0j} - \sum_{\mu=0}^\infty\hat{c}_{\mu,j-1}\sigma_k^{\mu+1}\cdot\delta_{i0} + \sum_{\mu,\nu\geq 0}\sigma_k^{\mu+1}\tau_l^{\nu+1}\cdot\delta_{i0}\delta_{0j}\right)_{i,j\geq 0}$$

$$= \left(\hat{c}_{i-1,j-1} - \hat{\gamma}_{i-1|:}\cdot\delta_{0j} - \hat{\gamma}_{:|j-1}\cdot\delta_{i0} + \hat{\gamma}_{:|:}\cdot\delta_{i0}\delta_{0j}\right)_{i,j\geq 0}.$$

This implies (42) and hence concludes the overall proof of the proposition. $\qquad\square$

### A.6 Proof of Proposition 2.8

*Proof.* We combine Lemma A.2 with a similar reasoning as for Proposition 2.7.
Proceeding by induction on $\eta := k + l$, we will show that, uniformly in $(s,t) \in T_{k,l}$,

$$\kappa_{k,l}(s,\tau_l) = \sum_{r=0}^\infty\alpha_r^{(k,l)}(s-\sigma_k)^r \quad\text{and}\quad \kappa_{k,l}(\sigma_k,t) = \sum_{r=0}^\infty\beta_r^{(k,l)}(t-\tau_l)^r \tag{43}$$

for each $k,l \in \{1,\ldots,\ell-1\}$ (which, in particular, implies that $\alpha_0^{(k,l)}(=\kappa_{k,l}(\sigma_k,\tau_l)) = \beta_0^{(k,l)}$), and use this to then prove (24). Starting with the base case $\eta = 2$, note that $(43)|_{(k,l)=(1,1)}$ is immediate from the boundary conditions in (2) and the definition of $\alpha_0^{(1,1)}$ and $\beta_0^{(1,1)}$. Thus, for $(k,l) = (1,1)$,

$$\kappa_{k,l} = \sum_{n=0}^\infty\left[\sum_{i=0}^\infty\tilde{\alpha}_i^{(k,l)}\boldsymbol{T}_{k,l}^n(\varphi_{\sigma_k}^{(i)}) + \sum_{j=0}^\infty\beta_j^{(k,l)}\boldsymbol{T}_{k,l}^n(\varphi_{\tau_l}^{(j)})\right] \tag{44}$$

by combination of (43) with Proposition 2.5, which holds uniformly on $T_{k,l}$ and for the zero-starting sequence $\big(\tilde{\alpha}_i^{(k,l)}\big)_{i \geq 0} := \big((1 - \delta_{i,0})\alpha_i^{(k,l)}\big)_{i \geq 0}$. Denoting $u_n := \sum_{i=0}^{\infty} \tilde{\alpha}_i \boldsymbol{T}_{k,l}^n(\varphi_{\sigma_k}^{(i)}) + \sum_{j=0}^{\infty} \beta_j \boldsymbol{T}_{k,l}^n(\varphi_{\tau_l}^{(j)})$ for each $n \in \mathbb{N}_0$, we obtain that, pointwise for each $(s,t) \in T_{k,l}$,

$$
u_{n+1}(s,t) = (\boldsymbol{T}_{k,l}(u_n))(s,t) = \rho_{k,l} \sum_{i,j=0}^{\infty} \hat{c}_{ij}^{(k,l)} \int_{\sigma_k}^s (\tilde{s} - \sigma_k)^i \, \mathrm{d}\tilde{s} \int_{\tau_l}^t (\tilde{t} - \tau_l)^j \, \mathrm{d}\tilde{t}
$$

$$
= \sum_{i,j=0}^{\infty} (s - \sigma_k)^{i+1} \frac{\rho_{k,l} \hat{c}_{ij}^{(k,l;n)}}{(i+1)(j+1)} (t - \tau_l)^{j+1} = \Big\langle \eta(s - \sigma_k), \big(\rho_{k,l} \underline{S} \hat{C}_n^{k,l} \underline{T}\big) \eta(t - \tau_l) \Big\rangle,
$$

(45)

where from the second equality onwards we assumed that $u_n = \langle \eta(\cdot - \sigma_k), \hat{C}_n^{k,l} \eta(\cdot - \tau_l) \rangle$ for some $\hat{C}_n^{k,l} \equiv (\hat{c}_{ij}^{(k,l;n)}) \in \ell_1(\mathbb{N}_0^2)$ (induction hypothesis). Since indeed

$$
u_0 = \sum_{i=0}^{\infty} \tilde{\alpha}_i^{(k,l)} (s - \sigma_k)^i + \beta_i^{(k,l)} (t - \tau_l)^i = \langle \eta(\cdot - \sigma_k), \hat{C}_0^{k,l} \eta(\cdot - \tau_l) \rangle
$$

(46)

for $\hat{C}_0^{k,l} := \big(\alpha_i^{(k,l)} \delta_{i0} + \beta_j^{(k,l)} \delta_{0j} - \kappa_{k-1,l-1}(\sigma_k, \tau_l) \delta_{0i} \cdot \delta_{0j}\big)_{i,j \geq 0}$, combining (46),(45), (44) proves

$$
\kappa_{k,l} = \Big\langle \eta(\cdot - \sigma_k), \hat{C}^{k,l} \eta(\cdot - \tau_l) \Big\rangle \Big|_{T_{k,l}} \quad \text{for} \quad \hat{C}^{k,l} := \sum_{n=0}^{\infty} \rho_{k,l}^n \underline{S}^n \hat{C}_0^{k,l} \underline{T}^n,
$$

(47)

by induction on $n \in \mathbb{N}_0$. (Note that at this point, (47) is established for $(k,l) = (1,1)$ only.)

Since $(\underline{S}^n)_{i,j} = \frac{j! \delta_{i,j+n}}{(j+n)!}$ and $(\underline{T}^n)_{i,j} = \frac{i! \delta_{j,i+n}}{(i+n)!}$ for each $i,j \geq 0$ and $n \in \mathbb{N}_0$ by definition of $\underline{S}, \underline{T}$,

$$
\big(\underline{S}^n \hat{C}_0^{k,l} \underline{T}^n\big)_{i,j \geq 0} = \begin{cases} \frac{\beta_{j-i}^{(k,l)}(j-n)!}{n! j!}, & i = n \text{ and } j \geq n, \\[2mm] \frac{\alpha_{i-j}^{(k,l)}(i-n)!}{i! n!}, & i \geq n \text{ and } j = n, \\[2mm] 0, & \text{else.} \end{cases}
$$

Consequently, the matrix entries of $\hat{C}^{k,l} =: (\hat{c}_{ij}^{(k,l)})_{i,j \geq 0}$ each read

$$
\hat{c}_{i,j}^{(k,l)} = \left\{ \begin{array}{ll} \frac{\rho_{k,l}^{\min(i,j)} \beta_{j-i}^{(k,l)} (j - \min(i,j))!}{\min(i,j)! j!}, & j \geq \min(i,j) \\[2mm] \frac{\rho_{k,l}^{\min(i,j)} \alpha_{i-j}^{(k,l)} (i - \min(i,j))!}{i! \min(i,j)!}, & i \geq \min(i,j) \end{array} \right\} = \big(A_{k,l} \odot B_{k,l} \odot W\big)_{i,j},
$$

(48)

implying $\hat{C}^{k,l} = \tilde{C}^{k,l}$, as desired. But then, in particular – recalling (47) and the definitions (26) –

$$
\kappa_{k,l+1}(s, \tau_{l+1}) = \big\langle \eta(s - \sigma_k), \hat{C}^{k,l} \eta(\tau_{l+1} - \tau_l) \big\rangle
$$

$$
= \big\langle \eta(s - \sigma_k), (\alpha_r^{(k,l+1)})_{r \geq 0} \big\rangle_{\ell^2(\mathbb{N})} = \sum_{r=0}^{\infty} \alpha_r^{(k,l+1)} (s - \sigma_k)^r, \quad \text{and analogously}
$$

$$
\kappa_{k+1,l}(\sigma_{k+1}, t) = \big\langle (\beta_r^{(k+1,l)})_{r \geq 0}, \eta(t - \tau_l) \big\rangle_{\ell^2(\mathbb{N})} = \sum_{r=0}^{\infty} \beta_r^{(k+1,l)} (t - \tau_l)^r,
$$

uniformly in $(s,t) \in T_{k,l}$. This—together with the fact that $\kappa_{k,1}(s,\tau_1) = 1$ $(= \sum_{r=0}^{\infty} \alpha_r^{(k,1)}(s - \sigma_k)^r$ for $(\alpha_r^{(k,1)})_{r \geq 0} := (\delta_{0,r})_{r \geq 0}$ and all $s \in [\sigma_k, \sigma_{k+1}]$, for each $k = 1, \ldots, \ell - 1$) and $\kappa_{1,l}(\sigma_1, t) = 1$ $(= \sum_{r=0}^{\infty} \beta_r^{(1,l)}(t - \tau_1)^r$ for $(\beta_r^{(1,l)})_{r \geq 0} := (\delta_{0,r})_{r \geq 0}$ and all $t \in [0,1]$, for each $l = 1, \ldots, \ell - 1)$—implies that (43) holds also for $\eta = 3$. In fact, the above argumentation—as stated—proves that if (43) holds for some fixed $k, l \in \{1, \ldots, \ell - 1\}$, then both (47) and (48) hold for this $(k, l)$ and also (provided $\max(k, l) \leq \ell - 2$) that (43) holds for $(k + 1, l)$ and $(k, l + 1)$. This proves the proposition. $\square$

## A.7 Gram Matrix Approximation Error

The remarks on local truncation error in Section C.3 can be directly extended to yield an explicit prior bound on the approximation error for the Gram matrix of a given family of time series: the double-sum structure of the local approximation error (59) allows for direct control through modified Bessel tails, which can then be scaled from component-wise to matrix-level bounds.

Note that an estimate very similar to the one underlying the following proof was first presented in [5].

**Proposition A.3.** *Let* $G \equiv (G_{ij}) := \big(K_{\boldsymbol{x}^{(i)}, \boldsymbol{x}^{(j)}}(1,1)\big)_{i,j=1,\ldots,m}$ *be the signature-kernel Gram matrix for a family* $\mathcal{X} \equiv \big(\boldsymbol{x}^{(i)} \equiv (\boldsymbol{x}_l^{(i)})_{l=1,\ldots,\ell} \mid i = 1,\ldots,m\big)$ *of time series in* $\mathbb{R}^d$. *Let further* $\hat{G}_N := \big(\hat{\kappa}_{\ell-1,\ell-1;\,\boldsymbol{x}^{(i)},\boldsymbol{x}^{(j)}}^{[N]}(1,1)\big)_{i,j=1,\ldots,m}$, *where* $\hat{\kappa}_{\ell-1,\ell-1}^{[N]}$ *is defined as in (24) but for the matrices*

$$\tilde{C}_N^{k,l} := A_{k,l}^{[N]} \odot B_{k,l}^{[N]} \odot W_N \qquad \big(k,l \in \{1,\ldots,\ell-1\}\big)$$

*with* $A_{k,l}^{[N]} := \big(\rho_{k,l}^{\min(i,j)}\big)_{0 \le i,j \le N}$ *and* $W_N := (w_{i,j})_{0 \le i,j \le N}$ *and* $B_{k,l}^{[N]} \equiv \big(b_{i,j}^{(k,l)|N}\big)_{0 \le i,j \le N}$ *for*

$$b_{i,i+r}^{(k,l)|N} := \alpha_r^{(k,l)|N} \qquad and \qquad b_{i,i-r}^{(k,l)|N} := \beta_{|r|}^{(k,l)|N}, \qquad for\ each\ (i,r) \in \mathbb{N}_0^2\,,$$

*where* $\beta_r^{(1,1)|N} = \alpha_r^{(1,1)|N} := \delta_{0,r}$ *for each* $r \in \mathbb{N}_0$ *and, recursively (recalling (26) for notation),*

$$\big(\alpha_r^{(k,l)|N}\big)_{r=0}^N := \tilde{C}_N^{k,(l-1)} \cdot \eta(1/(\ell-1)) \qquad and \qquad \big(\beta_r^{(k,l)|N}\big)_{r=0}^N := \big[\tilde{C}_N^{(k-1),l}\big]^\dagger \cdot \eta(1/(\ell-1)),$$

*for each* $k,l \in \{1,\ldots,\ell-1\}$. *Then we have the (truncation induced) approximation error estimate*

$$\big\|G - \hat{G}_N\big\| \le \gamma_{m,\|\mathcal{X}\|_\infty} \frac{\|\mathcal{X}\|_\infty^{N+1}\zeta_N}{(N+1)!^2}, \tag{49}$$

*with* $\|\cdot\|$ *the Frobenius norm,* $\|\mathcal{X}\|_\infty := \max_{1 \le i,j \le m;\, 1 \le k,l \le \ell-1} \big|\Delta_k \boldsymbol{x}^{(i)} \Delta_l \boldsymbol{x}^{(j)}\big|$, *and with*

$$\gamma_{m,\|\mathcal{X}\|_\infty} := \frac{m}{2} \prod_{\nu=0}^{2\ell-2} I_0\bigg(\frac{2\sqrt{\nu\|\mathcal{X}\|_\infty}}{\ell-1}\bigg) \quad and \quad \zeta_N := \Big[1 + \frac{2\ell-2}{N+2}\Big]\Big(\frac{2}{\ell-1}\Big)^{N+1}.$$

*Proof.* We proceed along the aforementioned lines, starting with the trivial norm inequality

$$\big\|G - \hat{G}_N\big\| \le m \max_{1 \le i,j \le m} \varepsilon_{ij}^{[N]}, \quad \text{for} \quad \varepsilon_{ij}^{[N]} := \big|K_{\boldsymbol{x}^{(i)}, \boldsymbol{x}^{(j)}}(1,1) - \hat{\kappa}_{\ell-1,\ell-1;\,\boldsymbol{x}^{(i)},\boldsymbol{x}^{(j)}}^{[N]}(1,1)\big|. \tag{50}$$

Now, for any fixed $(i,j) \in \{1,\ldots,m\}^{\times 2}$, the error $\varepsilon_{ij}^{[N]}$ reads, see (the last display of) Section A.6,

$$\varepsilon_{ij}^{[N]} = \big|\kappa_{\ell-1,\ell-1}(1,1) - \hat{\kappa}_{\ell-1,\ell-1}^{[N]}(1,1)\big| = \Bigg|\sum_{r=0}^\infty \xi_r^{\ell',\ell'}(1-\tau_{\ell-1})^r + \tilde{\xi}_r^{\ell',\ell'}(1-\sigma_{\ell-1})^r\Bigg| \tag{51}$$

for some coefficient sequences $\xi^{\ell',\ell'} \equiv (\xi_r^{\ell',\ell'})_{r \ge 0}$ and $\tilde{\xi}^{\ell',\ell'} \equiv (\tilde{\xi}_r^{\ell',\ell'})_{r \ge 0}$ (where we suppressed the dependence on $(i,j)$ and $N$ and denoted $\ell' := \ell - 1$ to ease notation). By linearity of the Goursat PDE (2) and its uniqueness-of-solution, we find by comparing coefficients (see Section A.6) that: $\xi^{\ell',\ell'} = \alpha^{\ell',\ell'} - \hat{\alpha}_N^{\ell',\ell'}$ and $\tilde{\xi}^{\ell',\ell'} = \beta^{\ell',\ell'} - \hat{\beta}_N^{\ell',\ell'}$ for the formerly defined coefficients $\alpha^{k,l} \equiv (\alpha_r^{(k,l)})_{r \ge 0}$ and $\hat{\alpha}_N^{k,l} \equiv (\alpha_r^{(k,l)|N})_{r \ge 0}$ and $\beta^{k,l} \equiv (\beta_r^{(k,l)})_{r \ge 0}$ and $\hat{\beta}_N^{k,l} \equiv (\beta_r^{(k,l)|N})_{r \ge 0}$, and that (as we recall)

$$(\alpha^{k,l}, \beta^{k,l}) = \mathfrak{A}_{k,l}(\alpha^{k,l-1}, \beta^{k-1,l}) \quad \text{and} \quad (\hat{\alpha}_N^{k,l}, \hat{\beta}_N^{k,l}) = \hat{\mathfrak{A}}_{k,l}(\hat{\alpha}_N^{k,l-1}, \hat{\beta}_N^{k-1,l})$$

for the (bounded) linear 'ADM-type' Goursat-solution operators $\mathfrak{A}_{k,l}, \hat{\mathfrak{A}}_{k,l} : \ell_1(\mathbb{N}_0)^{\times 2} \to \ell_1(\mathbb{N}_0)^{\times 2}$ defined (recursively) in Proposition 2.8 and (via projection $\ell_1(\mathbb{N}_0) \twoheadrightarrow \mathbb{R}^{N+1}$) Proposition A.3, respec-

tively. In this conceptualization, the coefficients $(\xi^{\ell',\ell'}, \tilde{\xi}^{\ell',\ell'})$ of the approximation error (51) read:

$$(\xi^{\ell',\ell'}, \tilde{\xi}^{\ell',\ell'}) = \mathfrak{A}_{\ell',\ell'}(\alpha^{\ell',\ell'-1}, \beta^{\ell'-1,\ell'}) - \hat{\mathfrak{A}}_{\ell',\ell'}(\hat{\alpha}_N^{\ell',\ell'-1}, \hat{\beta}_N^{\ell'-1,\ell'})$$

$$= \left(\pi_{(N,\infty)} \circ \mathfrak{A}_{\ell',\ell'}\right)(\alpha^{\ell',\ell'-1}, \beta^{\ell'-1,\ell'}) + \left[\left(\pi_{[0,N]} \circ \mathfrak{A}_{\ell',\ell'}\right)(\alpha^{\ell',\ell'-1}, \beta^{\ell'-1,\ell'}) - \hat{\mathfrak{A}}_{\ell',\ell'}(\hat{\alpha}_N^{\ell',\ell'-1}, \hat{\beta}_N^{\ell'-1,\ell'})\right]$$

$$= \left(\pi_{(N,\infty)} \circ \mathfrak{A}_{\ell',\ell'}\right)(\alpha^{\ell',\ell'-1}, \beta^{\ell'-1,\ell'}) + \left(\pi_{[0,N]} \circ \mathfrak{A}_{\ell',\ell'}\right)(\alpha^{\ell',\ell'-1} - \hat{\alpha}_N^{\ell',\ell'-1}, \beta^{\ell'-1,\ell'} - \hat{\beta}_N^{\ell'-1,\ell'})$$

$$= \left(\pi_{(N,\infty)} \circ \mathfrak{A}_{\ell',\ell'}\right)(\alpha^{\ell',\ell'-1}, \beta^{\ell'-1,\ell'}) + \left(\pi_{[0,N]} \circ \mathfrak{A}_{\ell',\ell'}\right)(\xi^{\ell',\ell'-1}, \tilde{\xi}^{\ell'-1,\ell'})$$

$$= \left(\pi_{(N,\infty)} \circ \mathfrak{A}_{\ell',\ell'}\right)(\alpha^{\ell',\ell'-1}, \beta^{\ell'-1,\ell'}) + \left(\pi_{[0,N]} \circ \mathfrak{A}_{\ell',\ell'}\right)\left[(\xi^{\ell',\ell'-1}, \tilde{\xi}^{\ell'-1,\ell'})^{\leq N}\right] \qquad (52)$$
$$+ \left(\pi_{[0,N]} \circ \mathfrak{A}_{\ell',\ell'}\right)\left[(\xi^{\ell',\ell'-1}, \tilde{\xi}^{\ell'-1,\ell'})^{>N}\right],$$

for the projection $\pi_{[0,N]} : \ell_1(\mathbb{N}_0)^{\times 2} \ni a \equiv \left(a_r^{(1)}, a_r^{(2)}\right)_{r\geq 0} \mapsto (a_r^{(1)} \mathbb{1}_{[0,N]}(r), a_r^{(2)} \mathbb{1}_{[0,N]}(r))_{r\geq 0} =: a^{\leq N} \in \ell_1(\mathbb{N}_0)^{\times 2}$ and its defect $\pi_{(N,\infty)} := \mathrm{id}_{\ell_1(\mathbb{N}_0)^{\times 2}} - \pi_{[0,N]}$, with $a^{>N} := \pi_{(N,\infty)}(a)$ for each $a \in \ell_1(\mathbb{N}_0)$. Note that the third of the above identities holds by the linearity of $\mathfrak{A}_{k,l}$ and since the operators $\pi_{[0,N]} \circ \mathfrak{A}_{k,l}$ and $\hat{\mathfrak{A}}_{k,l}$ coincide on the subspace $\pi_{[0,N]}(\ell_1(\mathbb{N}_0)^{\times 2})$.

Denoting the summands in (52) by $\varsigma^{(1)}, \varsigma^{(2)}$, and $\varsigma^{(3)}$, resp., direct (but tedious) computations show:

$$\max_{\nu=1,2} \left|\varsigma_r^{(1)|\nu}\right| \leq \varphi(2\ell'-1)\frac{(2\ell'|\mathfrak{x}_{ij}|)^r}{r!^2(\ell')^r}, \qquad (53)$$

$$\max_{\nu=1,2} \left|\varsigma_r^{(2)|\nu}\right| \leq I_0\left(\frac{2\sqrt{(2\ell'-1)|\mathfrak{x}_{ij}|}}{\ell-1}\right)\frac{(2\ell'|\mathfrak{x}_{ij}|)^r}{2(\ell')^r r!^2} \max_{\xi=\xi^{\ell',\ell'}, \tilde{\xi}^{\ell',\ell'};\, \tilde{r}\geq 0} \frac{\tilde{r}!^2(\ell')^{\tilde{r}}|\xi_{\tilde{r}}|}{((2\ell'-1)|\mathfrak{x}_{ij}|)^{\tilde{r}}}, \qquad (54)$$

$$\max_{\nu=1,2} \left|\varsigma_r^{(3)|\nu}\right| \leq \tilde{\eta}_{\ell',N}\, I_0\left(\frac{2\sqrt{(2\ell'-1)|\mathfrak{x}_{ij}|}}{\ell-1}\right)\frac{(2\ell'|\mathfrak{x}_{ij}|)^r}{2(\ell')^r r!^2} \max_{\xi=\alpha^{\ell',\ell'}, \beta^{\ell',\ell'};\, \tilde{r}\geq 0} \frac{\tilde{r}!^2(\ell')^{\tilde{r}}|\xi_{\tilde{r}}|}{((2\ell'-1)|\mathfrak{x}_{ij}|)^{\tilde{r}}}, \quad (55)$$

$$|\eta_r^{\ell',\ell'}| \leq \varphi(2\ell'-2)\frac{((2\ell'-1)|\mathfrak{x}_{ij}|)^r}{r!^2(\ell')^r} \qquad (\eta = \alpha, \beta) \qquad (56)$$

for each $r \in \mathbb{N}_0$, with $I_0 : x \mapsto I_0(x)$ the modified Bessel function of the first kind of order zero (cf. Lemma 2.4) and $|\mathfrak{x}_{ij}| := \max_{1\leq k,l\leq \ell-1} |\Delta_k \boldsymbol{x}^{(i)} \Delta_l \boldsymbol{x}^{(j)}|$, and for the function $\varphi(u) := \frac{1}{2}\prod_{\nu=0}^u I_0(2\sqrt{\nu|\mathfrak{x}_{ij}|}/\ell')$ and $\tilde{\eta}_{\ell',N} := \frac{(2\ell'-1)^{N+1}|\mathfrak{x}_{ij}|^{N+1}}{(\ell')^{(2N+2)}(N+1)!^2}$; see Sections A.5 and A.6 and cf. also [5, Props. 3.2, 3.3, 3.4]. Applying the estimates (53), (54), (55), (56) (on the size of the additive components $\varsigma^{(i)}$ in (52)) back to (51) via the triangle inequality and the auxiliary estimates

$$\left|\varsigma_r^{(2)} + \varsigma_r^{(3)}\right|_{1,1} \leq \sum_{\mu=2,3;\,\nu=1,2} \left|\varsigma_r^{(\mu)|\nu}\right| \leq \left[\theta_{\xi,\tilde{\xi}}\, I_0\left(\frac{2\sqrt{(2\ell'-1)|\mathfrak{x}_{ij}|}}{\ell-1}\right) + \varphi(2\ell'-1)\tilde{\eta}_{\ell',N}\right]\frac{(2\ell'|\mathfrak{x}_{ij}|)^r}{(\ell')^r r!^2},$$

$$\leq \varphi(2\ell'-1)\tilde{\tilde{\eta}}_{\ell',N}\frac{(2\ell'|\mathfrak{x}_{ij}|)^r}{(\ell')^r r!^2}\sum_{n=0}^{2\ell'-1} n^{N+1} \leq \varphi(2\ell'-1)\tilde{\tilde{\eta}}_{\ell',N}\frac{(2\ell')^{N+2}-1}{N+2}\frac{(2\ell'|\mathfrak{x}_{ij}|)^r}{(\ell')^r r!^2} \quad (r \in \mathbb{N}_0)$$
$$\tag{57}$$

with $\theta_{\xi,\tilde{\xi}} := \max_{\xi=\xi^{\ell',\ell'}, \tilde{\xi}^{\ell',\ell'};\, \tilde{r}\geq 0} \frac{\tilde{r}!^2(\ell')^{\tilde{r}}|\xi_{\tilde{r}}|}{((2\ell'-1)|\mathfrak{x}_{ij}|)^{\tilde{r}}}$ and $\tilde{\tilde{\eta}}_{\ell',N} := \frac{\tilde{\eta}_{\ell',N}}{(2\ell'-1)^{N+1}}$, then yields the error bound

$$\varepsilon_{ij}^{[N]} \leq \sum_{r=0}^N |\xi_r^{\ell',\ell'} + \tilde{\xi}_r^{\ell',\ell'}|(\ell')^{-r} + \sum_{r=N+1}^\infty |\xi_r^{\ell',\ell'} + \tilde{\xi}_r^{\ell',\ell'}|(\ell')^{-r} \qquad (58)$$

$$= \sum_{r=0}^N \left|\varsigma_r^{(2)} + \varsigma_r^{(3)}\right|_{1,1}(\ell')^{-r} + \sum_{r=N+1}^\infty \left|\varsigma_r^{(1)}\right|(\ell')^{-r}$$

$$\leq \varphi(2\ell'-1)\left[\tilde{\tilde{\eta}}_{\ell',N}\frac{(2\ell')^{N+2}-1}{N+2}\sum_{r=0}^N \frac{(2\ell'|\mathfrak{x}_{ij}|)^r}{(\ell')^{2r}r!^2} + \sum_{r=N+1}^\infty \frac{(2\ell'|\mathfrak{x}_{ij}|)^r}{(\ell')^{2r}r!^2}\right]$$

$$\leq \varphi(2\ell')\left[\frac{2^{N+2}|\mathfrak{x}_{ij}|^{N+1}}{(\ell')^N(N+1)!^2(N+2)} + \frac{2^{N+1}|\mathfrak{x}_{ij}|^{N+1}}{(\ell')^{N+1}(N+1)!^2}\right] = \varphi(2\ell')\zeta_N\frac{|\mathfrak{x}_{ij}|^{N+1}}{(N+1)!^2}.$$

Note that the second and third inequality in (57) follow from (54) and (55) via a straightforward induction, see also [5, Prop. 3.3]. The desired inequality (49) now follows immediately by combination of (58) and (50), using also that the function $[0, \infty) \ni u \mapsto I_0\left(\frac{2\sqrt{\nu u}}{\ell - 1}\right)$ is monotone. $\qquad\square$

# B  Benchmarking

## B.1  Downstream Experiments: Additional Figures

Complete implementation details and hyperparameter grids for the experiments underlying the following figures (see Section 3) are available in the `PowerSig` GitHub repository at

$$\texttt{https://github.com/geekbeast/powersig}$$

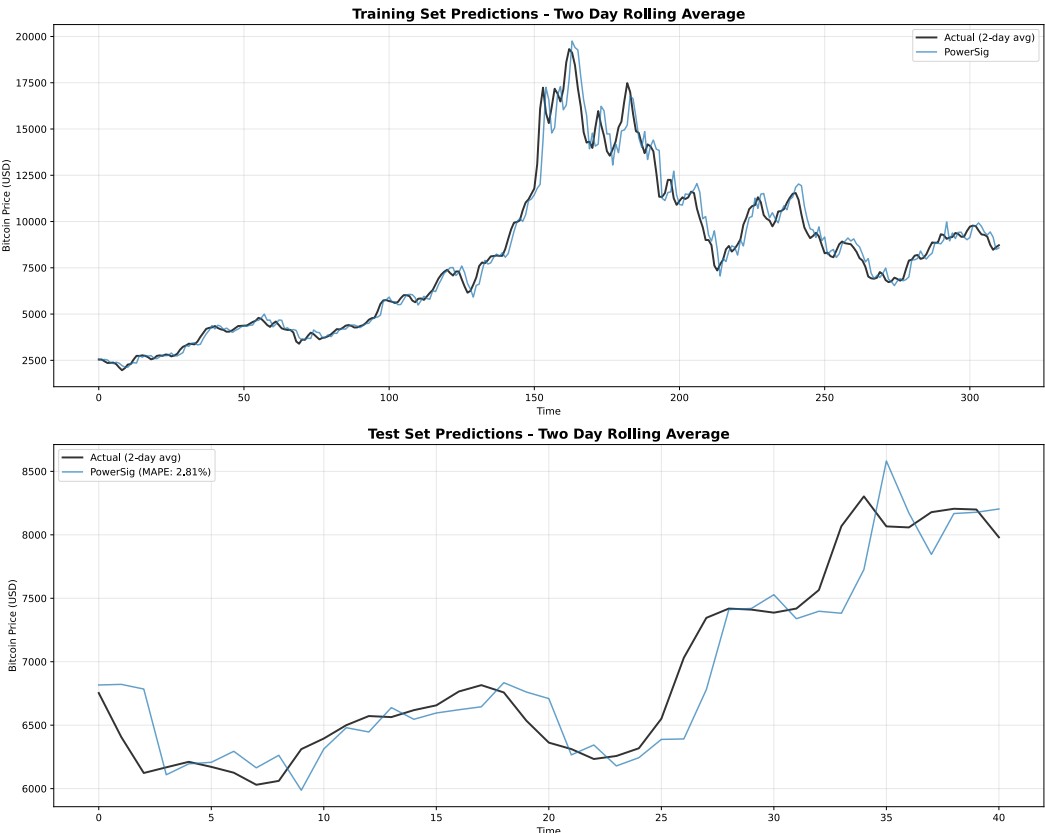

Figure 4: *Bitcoin price regression (two-day rolling average). Top: training fit; bottom: test fit.* `PowerSig` *attains* $2.81\%$ *MAPE (under the default linear static kernel) compared to* $3.23\%$ *MAPE for the RBF-assisted* `KSig-PDE`, *while using only a fraction of the device memory.*

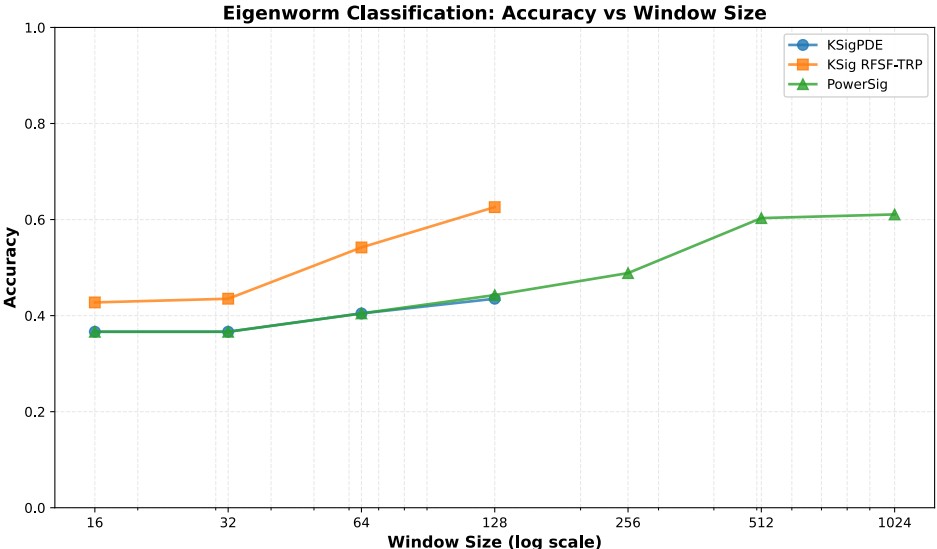

Figure 5: UEA Eigenworms classification across window lengths $L$. Test accuracy versus input window length $L$ for `PowerSig`, `KSig-PDE`, and the RFF baseline `RFSF-TRP`. `PowerSig` remains competitive and scales to $L = 1048$ with $61.1\%$ accuracy; `KSig-PDE` is competitive up to $L = 128$ before running out of memory (OOM). `RFSF-TRP` attains a slightly higher peak of $62.5\%$ at $L = 128$ but OOMs for larger $L$, consistent with the storage advantages in Fig. 3. This shows that extending the input window, enabled here at scale by `PowerSig`, can narrow performance gaps often ascribed to inductive bias while preserving feasibility.

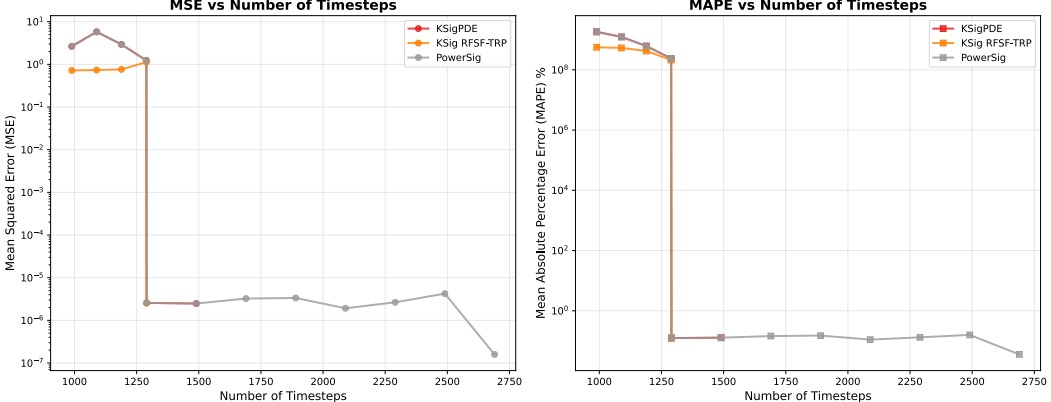

Figure 6: On long-horizon periodic signals, SVM-regression error versus input-window length for synthetic near-periodic time series with adjustable period. For representative instances, error decreases monotonically as windows span multiple periods. `PowerSig` sustains this favorable trend at window lengths beyond the reach of conventional or low-rank signature-kernel baselines, while maintaining peak memory that grows linearly with window length.

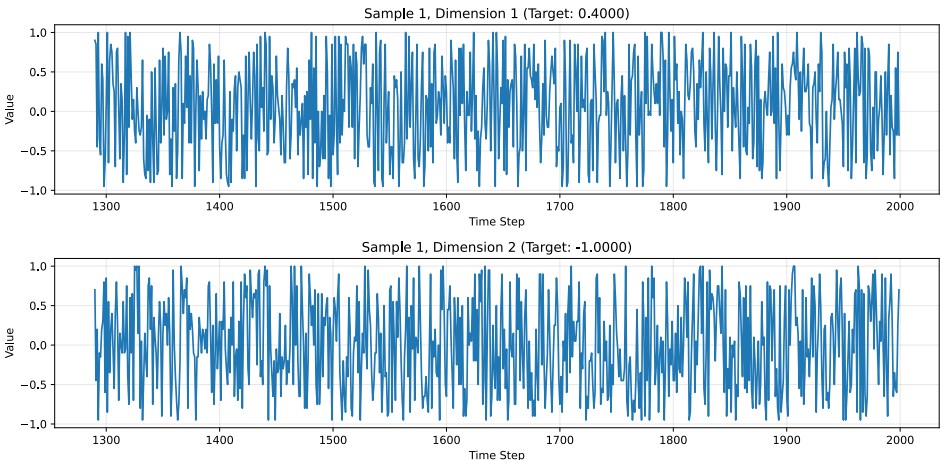

Figure 7: Representative near-periodic synthetic time series with adjustable period length (industrial/sensing proxy) used in the long-horizon experiments above. This instance illustrates the quasi-periodic structure across multiple cycles on which SVM-regression is evaluated; as input windows for SVM-regressors span several periods, their error decreases monotonically, with `PowerSig` sustaining this trend at longer windows while preserving linear peak-memory growth (Figure 6).

### B.2 Comparison Against `polysigkernel`

We benchmarked and contrasted our method (`PowerSig`) against the concurrent work `polysigkernel` from Cass et al. [5]. Both our approach and theirs were developed independently and released within two weeks of each other, with our preprint and theirs sharing the same arXiv month stamp.

Using the public JAX implementation of `polysigkernel`, we ran identical experiments on computation time and memory usage over two standard benchmarks: ($\alpha$) fixed-length ($= 512$) paths of increasing dimension, ranging from 2 to 4096, and ($\beta$) 2D Brownian motion paths of length 2 to 512. The results, shown in Figures 8 and 9, respectively, show that both solvers achieve essentially identical accuracy throughout while `PowerSig` performs better in runtime and memory usage. In benchmark ($\beta$), `PowerSig` was on average $3.1\%$ faster and required $23\%$ less memory than `polysigkernel` on oscillatory time series of length 512 (averaged over 10 iid samples). In benchmark ($\alpha$), `PowerSig` exhibited a comparable computational advantage, being $8.5\%$ faster on average at dimension 512, with similar memory usage.

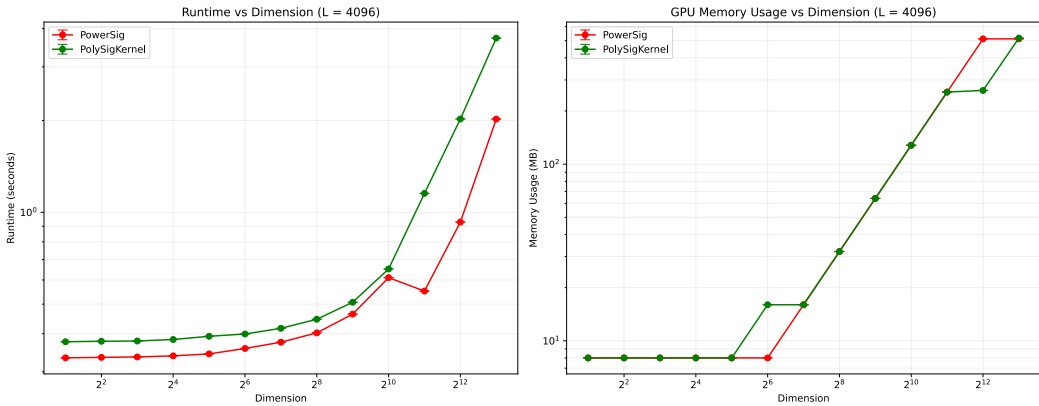

Figure 8: Scaling wrt. dimension (benchmark ($\alpha$)): Runtime and peak memory at fixed length $\ell = 512$ with dimension ranging from $d = 2$ to $d = 4096$. `PowerSig` is $8.5\%$ faster on average at dimension 512, with similar memory usage.

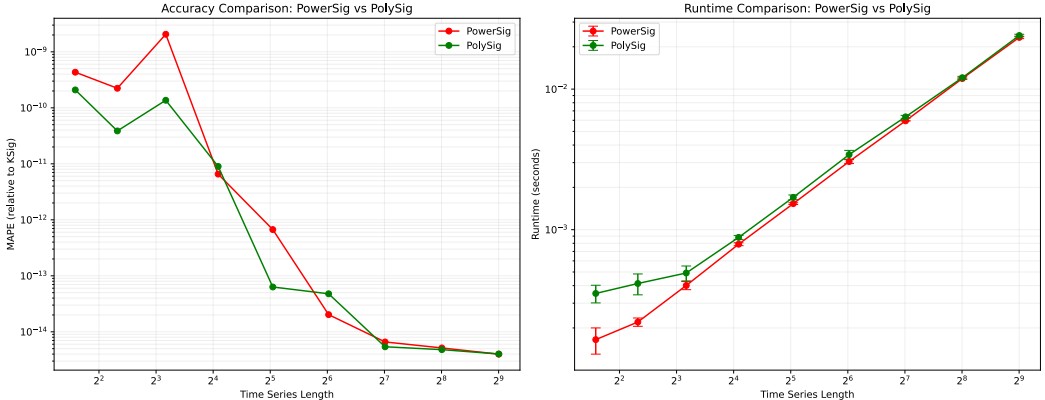

Figure 9: Scaling wrt. time series length (benchmark ($\beta$)): Runtime and peak memory on 2D Brownian paths with length ranging from $\ell = 2$ to $\ell = 512$ for `PowerSig` and `polysigkernel`. Accuracy is essentially identical, while `PowerSig` is $3.1\%$ faster on average and uses $23\%$ less memory at length $512$ (mean over $10$ iid samples).

Let us clarify why these empirical differences occur and how our underlying methods differ in concept despite propagating the same local power-series map: `polysigkernel` starts from an explicit Riemann-function formula to rewrite the Goursat solution as a sum of integrated modified Bessel functions $I_0$ [5, Thm. 3.1 (Polyanin)] which, after expanding each $I_0$ in its well-known power series, allows the kernel on every rectangle to be reduced (using integration-by-parts and a change of variables) to two univariate centred power series in $s$ and $t$, for which the resulting coefficient arrays can then be generated by the closed-form recurrence $\Lambda_{C;[\sigma_k, \sigma_{k+1}] \times [\tau_l, \tau_{l+1}]}$ in [5, Props. 3.1, A.1, A.2]. Truncating these expansions up to some fixed degree $N$ yields a recursive polynomial approximation scheme (`polysigkernel`) for which the authors provide local and global truncation error bounds [5, Prop. 3.4; Thm. 3.2]. (A lower-performant Chebyshev-interpolation scheme for polynomial boundary data is also discussed in [5, Section 3.3].) Our approach, by contrast, approaches the same PDE from a functional-analytic view: Using the Volterra formulation of the Goursat problem, we express the kernel propagator through an integral operator of spectral radius zero (Lemma 2.3) to obtain a concise, three-step Neumann recursion (Proposition 2.5) whose iterates build the same local power series coefficient-by-coefficient in a way that allows for their immediate coefficient extraction on the fly (Proposition 2.8). The resulting iteration adapts per tile and stops once the next term is below machine precision (typically 5–8 iterations; no global cut-off $N$ required), leading to a one-strip memory profile that is below the quadratic footprint of the reference `polysigkernel` implementation.

In summary, `polysigkernel` obtains each tile's coefficients from a closed-form Bessel (or Chebyshev) recurrence evaluated up to a user-chosen degree $N$, whereas `PowerSig` builds the same coefficients adaptively via a Neumann iteration that stops once the next term drops below machine precision. Both implement the same local propagation map and are analytically interchangeable per tile, with their different constructions yielding distinct practical profiles with a modest but consistent performance edge for `PowerSig` our matched head-to-head benchmarks (Figures 9 and 8).

## C   Summary of `PowerSig`: Methodology, Implementation, and Limitations

### C.1   Algorithm

The core idea behind our proposed signature kernel approximation method, `PowerSig`, is illustrated in Figure 1 of the main text. A summarizing description, in pseudocode, of an efficient Python implementation of this method—also used for our benchmarks—is provided in Figure 10 below.

### C.2   On Benchmarking

We benchmarked algorithms systematically in terms of runtime, memory usage, and accuracy. To ensure consistency and reproducibility, we developed a standardized benchmarking framework

**Algorithm: PowerSig**

1: **procedure** PROCESSDIAGONALS($X[0 .. cols-1]$, $Y[0 .. rows-1]$, $order$)
2:     Initialize $\mathbf{s}^{(0)}[0] \leftarrow [1, 0, \ldots, 0]$
3:     Initialize $\mathbf{t}^{(0)}[0] \leftarrow [1, 0, \ldots, 0]$
4:     **for** $d \leftarrow 0$ to $rows + cols - 3$ **do**
5:         $start\_row \leftarrow \max(0, d - (cols - 1))$
6:         $end\_row \leftarrow \min(d, rows - 1)$
7:         $L \leftarrow end\_row - start\_row + 1$
8:         Initialize $\mathbf{s}^{(d+1)}[0 .. L]$, $\mathbf{t}^{(d+1)}[0 .. L]$ as zero vectors
9:         **for** $k \leftarrow 0$ to $L - 1$ **do**
10:             $i \leftarrow start\_row + k$
11:             $j \leftarrow d - i$
12:             **if** $i + 1 < rows$ and $j + 1 < cols$ **then**
13:                 Compute $\rho \leftarrow \langle X[j+1] - X[j], Y[i+1] - Y[i] \rangle$
14:                 Form Toeplitz matrix $U$ from $\mathbf{t}^{(d)}[k]$ (first column) and $\mathbf{s}^{(d)}[k][1 :]$ (first row)
15:                 Form $R_{ij} \leftarrow \rho^{\min(i,j)}$ for $i, j = 0, \ldots, order$
16:                 $\mathbf{v}_{\text{col}} \leftarrow [\alpha^0, \ldots, \alpha^{\text{order}}]^T$,    $\alpha = \frac{1}{cols-1}$
17:                 $\mathbf{v}_{\text{row}} \leftarrow [\beta^0, \ldots, \beta^{\text{order}}]$,    $\beta = \frac{1}{rows-1}$
18:                 $\mathbf{s}_{\text{new}} \leftarrow \mathbf{v}_{\text{row}} \cdot (U \circ R)$
19:                 $\mathbf{t}_{\text{new}} \leftarrow (U \circ R) \cdot \mathbf{v}_{\text{col}}$
20:                 **if** $j < cols - 1$ **then**                                  ▷ Before right edge
21:                     **if** $k > 0$ **then**
22:                         $\mathbf{s}^{(d+1)}[k+1] \leftarrow \mathbf{s}_{\text{new}}$
23:                     **else**
24:                         $\mathbf{s}^{(d+1)}[0] \leftarrow [1, 0, \ldots, 0]$                 ▷ Initial condition
25:                     **end if**
26:                     $\mathbf{t}^{(d+1)}[k] \leftarrow \mathbf{t}_{\text{new}}$
27:                 **else**                                                     ▷ At or after right edge
28:                     $\mathbf{s}^{(d+1)}[k] \leftarrow \mathbf{s}_{\text{new}}$
29:                     **if** $k > 0$ **then**
30:                         $\mathbf{t}^{(d+1)}[k-1] \leftarrow \mathbf{t}_{\text{new}}$
31:                     **else**
32:                         $\mathbf{t}^{(d+1)}[k] \leftarrow [1, 0, \ldots, 0]$             ▷ Initial condition at far edge
33:                     **end if**
34:                 **end if**
35:             **end if**
36:         **end for**
37:     **end for**
38: **end procedure**

Figure 10: Our implementation of `PowerSig`, shown here, is based on the algorithmic approach that we developed in Section 2 and summarized in Figure 1.

comprising a base benchmarking class, a custom context manager, and a proxy wrapper of the CuPy allocator. While this setup directly supported PyTorch and CuPy, special considerations were necessary for JAX, as its XLA backend does not natively allow resetting peak memory usage between runs. We circumvented this limitation by isolating each JAX benchmarking run in a separate Python subprocess, ensuring accurate GPU memory measurements and preventing JAX's default behavior of pre-allocating 75% of GPU memory. Additionally, to prioritize numerical accuracy required by our experiments, we configured JAX explicitly to use 64-bit (f64) floating-point precision, overriding its default preference for computational speed.

### C.3  Error Analysis and Limitations

We identify two primary limitations of our proposed method for computing signature kernels: numerical stability (truncation errors) for very large $|\rho_{x,y}|$ (see also Proposition A.3 for details), and computational overhead due to JAX recompilation.

**Truncation Error**  Our computation of the signature kernel effectively involves linear operations to evaluate polynomial expansions, the latter given by the power series (24) for which by construction:

$$\kappa_{k,l}^{[N]}(s,t) := \sum_{i,j=0}^{N} \tilde{c}_{i,j}^{(k,l)}(s - \sigma_k)^i(t - \tau_l)^j \equiv \sum_{i,j=0}^{N} \frac{\rho_{k,l}^{\min(i,j)}(s - \sigma_k)^i(t - \tau_l)^j}{\prod_{\substack{\mu=1+j-\min(i,j) \\ \nu=1+i-\min(i,j)}}^{N} \mu\nu} \quad \big((s,t) \in T_{k,l}\big),$$

for any truncation order $N \in \mathbb{N}$. Evaluating this at the tile boundary point $(\sigma_{k+1}, \tau_{l+1})$ then yields:

$$\left\| \kappa_{k,l} - \kappa_{k,l}^{[N]} \right\|_{\infty;T_{k,l}} \lesssim \sum_{i,j=N+1}^{\infty} |\rho_{k,l}|^{\min(i,j)}(\Delta\sigma)^i(\Delta\tau)^j \tag{59}$$

for $\Delta\sigma := \sigma_{k+1} - \sigma_k$ and $\Delta\tau := \tau_{l+1} - \tau_l$. As $\Delta\sigma, \Delta\tau \to 0$ with increasing time series length, terms involving $(\Delta\sigma)^i$ and $(\Delta\tau)^j$ will rapidly fall below machine precision at a speed counterbalanced by the magnitude of the $|\rho_{k,l}|$ values; large values of $|\rho_{k,l}|$ significantly amplify the truncation errors (59), necessitating higher polynomial orders $N$ (which again are bounded by numerical precision).

However, truncation errors are typically localized, decaying rapidly across tiles, which helps to limit the overall impact of a few tiles with higher error on the accuracy of the final solution. To further mitigate truncation issues, our implementation configures JAX to use 64-bit floating-point precision (f64) and includes a mechanism for estimating the minimal required truncation order $N$ to meet a specified truncation error tolerance. Roughly speaking, we evaluate the intermediate ADM coefficient matrix for the maximum $|\rho_{k,l}|$ in the grid and then compute the sum of all entries in the last column, scaled by $\Delta\sigma^N$, and in the last row, scaled by $\Delta\tau^N$. This provides a bound on the truncation error and allows for a straightforward search over $N = 8, \ldots, 64$ to identify a suitable truncation order for a given set of kernel-constituting time series.

A more detailed discussion of approximation errors, including a rigorous analysis of the local truncation errors (59), is provided in Section A.7.

**JAX Recompilation**  JAX's compilation mechanism can incur significant overhead when processing batches of time series with varying lengths ("jagged" datasets). One potential mitigation strategy—using re-interpolating to pad each series to a uniform length—is not ideal, as it may distort the underlying temporal spacing and thus affect signature kernel computations. However, most datasets are trimmed and chunked to the same length so this is typically a minor limitation as long as cardinality of the set of input shapes is not approximately the same as the number of time series being compared.

