# OpenReview forum: "Scalable Signature Kernel Computations via Local Neumann Series Expansions"
_NeurIPS.cc/2025/Conference — NeurIPS 2025 poster_

### Official Review · Reviewer_4Qtm · 2025-06-29

**Clarity:** 3
**Significance:** 2
**Originality:** 2
**Rating:** 4
**Confidence:** 3

**Summary:**

This paper introduces PowerSig, a new method for computing the signature kernel—a powerful but computationally expensive kernel function for comparing time series based on rough path theory. The key idea is to avoid the traditional heavy global computation (e.g., solving a dense PDE or using dynamic programming) and instead break the kernel domain into small tiles, then compute local power series approximations within each tile using Neumann series expansions. Each tile’s kernel value is computed recursively using only its immediate neighbors, which significantly reduces memory usage and makes the method highly parallelizable. Because the power series converge quickly and can be adaptively truncated, the method maintains high accuracy even on very long or rough time series. The authors demonstrate strong empirical improvements in speed, memory efficiency, and numerical stability over existing state-of-the-art solvers.

**Questions:**

Do you have any theoretical insights or empirical heuristics that guide truncation depth per tile (e.g., based on local values of \rho)?

**Ethical Concerns:**

["NO or VERY MINOR ethics concerns only"]

**Final Justification:**

The authors have successfully addresses my major concerns, i.e. benchmarking with a similar methods, providing more guidance on how to choose truncation depth, etc. I will be happy to see all these being incorporated with the camera-ready version, which makes the paper more compelling and comprehensive.

**Limitations:**

While the results for low-Hurst fractional Brownian motion are impressive, I wonder about more pathological edge cases. Are there regimes (e.g., extremely high-dimensional paths or adversarial oscillations in x and y) where the power series becomes unstable or requires very high truncation orders?

**Paper Formatting Concerns:**

No concern

**Quality:**

3

**Strengths And Weaknesses:**

I found the technical contribution practical. The use of localized Neumann series to approximate the signature kernel is a smart way to circumvent the global memory bottleneck, and the paper demonstrates that this leads to substantial improvements in scalability. The method is grounded in solid math (with rigorous derivations and proofs), and the experimental results back up the claims well -- especially the ability to handle time series with over a million points on a single GPU. I also appreciate how the algorithm leverages the geometry of the time series in a structured, recursive way. This feels like a well-engineered solution to a real bottleneck in signature kernel applications. One area I wish the paper had gone a bit deeper is in the comparison and discussion with related or emerging alternatives. For example, the concurrent work by Cass et al. is mentioned but not actually benchmarked or contrasted. I feel like this is a critical piece for the completeness of this work. It would help contextualize PowerSig’s practical impact more clearly if the authors could situate it relative to other fast approximations or sparse kernel techniques (beyond KSig). Also, while the math is clean, some of the technical exposition -- particularly in Section 2.2 -- feels heavy and could benefit from more intuitive scaffolding or diagrams to make the recursion scheme easier to follow on a first read.

---

> ### Author Rebuttal · Authors · 2025-07-31
>
> Thank you very much for your time and the helpful comments and questions. In what follows, we address each point separately, hopefully answering any concerns you may have. Kindly let us know if you wish any further clarifications or answers.
>
> 1. One area I wish the paper had gone a bit deeper is in the comparison and discussion with related or emerging alternatives. For example, the concurrent work by Cass et al. is mentioned but not actually benchmarked or contrasted. I feel like this is a critical piece for the completeness of this work.
>
> Reply: We have since benchmarked and contrasted our method against the concurrent work of Cass et al. Both approaches were developed independently and released within two weeks of each other, with both preprints carrying the same arXiv month stamp. To address the understandable request for a direct comparison of our methods, our revised manuscript now contains a new subsection entitled "Benchmark against polysigkernel", accompanied by an additional figure panel showcasing performance on joint benchmarks. Using the publicly available JAX implementation of $\texttt{polysigkernel}$, we ran identical experiments on computation time and memory usage over two standard benchmarks: $(\alpha)$ 2D Brownian motion paths of length 2 to 512, and $(\beta)$ fixed-length (= 512) paths of increasing dimension, ranging from 2 to 4096. These experiments, which we are happy to detail further upon request and will include with the camera-ready version, show that both solvers achieve essentially identical accuracy throughout while $\texttt{PowerSig}$ performs better in runtime and memory usage. In benchmark $(\alpha)$, $\texttt{PowerSig}$ was on average 3.1% faster and required 23% less memory than $\texttt{polysigkernel}$ on oscillatory time series of length 512 (averaged over 10 iid samples). In benchmark $(\beta)$, $\texttt{PowerSig}$ exhibited a comparable computational advantage, being 8.5% faster on average at dimension 512, with similar memory usage.
>
> Our added subsection also clarifies why these empirical differences occur and how our underlying methods differ in concept despite propagating the same local power‑series map: $\texttt{Polysigkernel}$ starts from an explicit Riemann‑function formula to rewrite the Goursat solution as a sum of integrated modified Bessel functions $I_{0}$ [Cass et al., Thm. 3.1 (Polyanin)] which, after expanding each $I_{0}$ in its well‑known power series, allows the kernel on every rectangle to be reduced (using integration‑by‑parts and a change of variables) to two univariate centred power series in $s$ and $t$, for which the resulting coefficient arrays can then be generated by the closed‑form recurrence $\Lambda_{C;[\sigma_{k},\sigma_{k+1}]\times[\tau_{l},\tau_{l+1}]}$ (their Props. 3.1, A.1, A.2). Truncating these expansions up to some fixed degree $N$ yields a recursive polynomial approximation scheme ($\texttt{polysigkernel}$) for which the authors provide local and global truncation error bounds (their Prop. 3.4 and Thm. 3.2). A lower-performant Chebyshev‑interpolation scheme for polynomial boundary data is also discussed in their Sect. 3.3. Our algorithm, by contrast, approaches the same PDE from a functional-analytic view: Using the Volterra formulation of the Goursat problem, we express the kernel propagator through an integral operator with favourable spectrum (Lemma 2.3) that permits a concise three-step Neumann recursion (Prop. 2.5) whose iterates build the same local power series coefficient-by-coefficient in a way that allows for their immediate extraction on the fly (Prop. 2.8). This Neumann expansion terminates as soon as the next term is below machine precision, so iteration depth that can vary per tile and no global cut‑off $N$ is required; in practice 5–8 iterations suffice except on the roughest tiles, resulting in a memory footprint that stores only one tile strip and is strictly lower than the quadratic complexity held by the given $\texttt{polysigkernel}$ implementation.
>
> In summary, $\texttt{polysigkernel}$ obtains each tile's coefficients from a closed‑form Bessel (or Chebyshev) recurrence evaluated up to a user‑chosen degree $N$, whereas $\texttt{PowerSig}$ builds the same coefficients adaptively via a Neumann iteration that stops once the next term drops below machine precision. Because both procedures implement exactly the same propagation map, they are analytically interchangeable on every tile.  Their distinct constructions, however, naturally lead to different implementations and practical profiles, and our head‑to‑head benchmarks show a modest but consistent performance edge for $\texttt{PowerSig}$.
>
> 2. It would help contextualize $\texttt{PowerSig}$'s practical impact more clearly if the authors could situate it relative to other fast approximations or sparse kernel techniques (beyond KSig).
>
> Reply: Thank you for this suggestion. We agree and are currently working to implement it by benchmarking $\texttt{PowerSig}$ against the recent specialised low-rank and RFF-based approaches introduced in [Tóth, Oberhauser, and Szabó: "Random Fourier Signature Features", SIAM J. Math. Data Sci., Vol. 7 (1)]. We expect to have results available in the coming days and will include them in the camera-ready version. Any additional suggestions of specific sparse kernel techniques or alternative fast approximations suitable for comparison are welcome.
>
> 3. Also, while the math is clean, some of the technical exposition -- particularly in Section 2.2 -- feels heavy and could benefit from more intuitive scaffolding or diagrams to make the recursion scheme easier to follow on a first read.
>
> Reply: Thank you for pointing this out. We share your concern and will address this in two ways in the camera-ready version: a) the summarising schematic Figure 3 (currently in Appendix B), which illustrates our recursion scheme and its information flow, will be moved to the main text and placed alongside the current summary paragraph at the start of Section 2.2. We will also expand the figure caption to walk the reader through a full propagation step, clarifying the role of each object before any algebra appears. b) Additionally, we will add a short introductory paragraph (four to five sentences) that informally explains why only the south‑ and west‑edge data of a tile enter the update, thereby linking the figure more explicitly to the characteristic structure of the Goursat PDE. With these adjustments, the reader will first encounter an intuitive scaffold before engaging with the formal derivation, while all mathematical information remains unchanged.
>
> 4. Do you have any theoretical insights or empirical heuristics that guide truncation depth per tile (e.g., based on local values of $\rho$)?
>
> Reply: Yes, we do. Proposition 2.8 (cf. also Appendix B.3) shows that the approximation error of our expansion on a tile of side-length $\Delta$ at truncation depth $N$ is bounded by $\sum_{i,j \,:\, \max(i,j)\geq N+1}a_{ij}$ for $a_{ij}:=(|\rho|\Delta^2)^{\min(i,j)}/(\min(i,j)!)^2$, where $\rho$ is the local value of the Goursat PDE coefficient function on that tile. As described in the appendix, we have taken this inequality as the basis for a utility function, implemented as estimate_required_order() in the provided file powersig.jax.algorithm.py, which takes as input time series $(x_\mu \mid \mu = 1,\ldots, m)$ and $(y_\nu \mid \nu = 1,\ldots, n)$, along with a desired error tolerance, and suggests an optimal global truncation depth (capped at 64) for computing the cross-Gram matrix $(K_{x_\mu,y_\nu})\in\mathbb{R}^{m\times n}$. In our use cases, we found the numerical sweet spot to lie between truncation orders 7 and 15; beyond this range, rescaling the time series, rather than increased truncation depth, was the more effective way to restore stability.
>
> 5. While the results for low-Hurst fractional Brownian motion are impressive, I wonder about more pathological edge cases. Are there regimes (e.g., extremely high-dimensional paths or adversarial oscillations in x and y) where the power series becomes unstable or requires very high truncation orders?
>
> Reply: This points to an important aspect, thank you for raising it. As discussed in Appendix B.3, the numerical efficiency and stability of our tile-based approximation are closely tied to the behaviour of the inner product coefficient $\rho_{x,y}(s,t) =\langle \frac{dx}{ds},\frac{dy}{dt} \rangle$ that governs the local Goursat PDE. Instabilities can occur when $\rho_{x,y}$ exhibits large spikes on isolated tiles, pushing local truncation orders high. This typically arises when the time series have strongly heterogeneous increments, e.g. a few exceptionally large differences amid otherwise small ones, leading to extreme variations in $|\rho_{x,y}|$ across tiles. Such behaviour can break the 'generic situation' that local power series decay rapidly and uniformly.
>
> Adversarial oscillations (in the sense of irregular or bursty increments) can trigger this, particularly when isolated increments dominate in magnitude. High-dimensional paths, by contrast, are usually benign unless they exhibit the same large-increment structure, since the dimensionality enters only through $\rho_{x,y}$. In practice, these edge cases can be flagged when our internal utility function falls outside the typical truncation range of $[7,15]$. We have found that stability is then often recoverable via simple, invertible preprocessing (e.g., normalization or whitening) to compress the range of $|\rho_{x,y}|$ into a numerically stable regime.

---

> > ### Comment · Reviewer_4Qtm · 2025-08-03
> > **Response to rebuttal**
> >
> > I would like to thank the authors to the extra work for addressing my concerns, especially for accommodating the "benchmark request" for the new algorithm proposed within two weeks of release. I will consider raising my ratings as long as the points above are incorporated in the camera-ready version.

---

> > > ### Author Response · Authors · 2025-08-08
> > >
> > > We would like to thank the referee sincerely for their comments and for acknowledging our rebuttal. We confirm that all points raised in our response will be fully incorporated into the camera-ready version of the paper.
> > >
> > > Since submitting our rebuttal, we have further confirmed and consolidated additional empirical evidence of our approach’s strong real-world utility. In particular, we can now report concrete use cases where $\texttt{PowerSig}$'s ability to efficiently process long time series leads to genuine accuracy gains in downstream regression and classification tasks that were previously unachievable with conventional or low-rank-based signature kernel methods. These new findings, which are detailed in our comment above, will also be incorporated into the camera-ready version.
> > >
> > > We remain happy to address any remaining questions or comments the reviewer may have. Otherwise, we would like to thank the reviewer once again for their time, valuable constructive feedback, and consideration.

---

### Official Review · Reviewer_p9wB · 2025-07-03

**Clarity:** 2
**Significance:** 3
**Originality:** 3
**Rating:** 5
**Confidence:** 2

**Summary:**

This paper presents PowerSig, a space-efficient approach to compute signature kernels. The signature kernel is a tool to analyze time series, which was previously generally computed via dynamic programming or as a solution to a Goursat PDE problem. Instead of solving a global PDE problem, this paper proposes to partition the unit square and solve many local PDE problems instead. These are reformulated as integral equations which admit local power series expansion solutions. Coefficients are first computed for the tile $T_{1,1}$ in the corner, after which boundary conditions are expanded alongside diagonals $T_{i, j: i+j=C}$ until the opposite corner tile is reached. The proposed method has similar time complexity compared to existing approaches, but requires significantly less memory especially for long sequences.

**Questions:**

1. Could the algorithm be modified to start at any of the four corner tiles? If not, why? If yes, could you simultaneously start from every corner and process tiles in parallel towards the middle of the square?

2. Can you combine aspects of both your method and polysigkernel, or are they incompatible with each other?

*Minor comments/typos*

L56: "solition"

Section 2.1: Why use a notation with 2 different "l" symbols ($\ell, l$)?

**Ethical Concerns:**

["NO or VERY MINOR ethics concerns only"]

**Final Justification:**

This submission introduces a space-efficient approach to compute signature kernels. In their rebuttal, the authors addressed the clarity issue I raised and started applying their method to more complex scenarios.

**Limitations:**

Yes

**Quality:**

4

**Strengths And Weaknesses:**

**Strengths**

- The proposed approach, due to its lower space complexity, allows to efficiently scale signature kernel computations to very long or rough time series, which is validated by numerical experiments.

- The rapid convergence of the power series leads to highly accurate results, which may surpass existing PDE-based methods.

- I find the overall approach elegant as it converts a complex global problem into simpler local ones.

**Weaknesses**

- For someone not already closely familiar with the signature kernel (in particular as a solution to a Goursat PDE), the mathematical derivations can be difficult to follow.

- To further demonstrate the usefulness of the proposed method and reinforce the claim that it is suitable for large-scale applications in different domains, it could have been interesting to show how it could allow solving a previously intractable real-world problem.

---

> ### Author Rebuttal · Authors · 2025-07-31
>
> Thank you very much for your time and your very valuable feedback and suggestions. In what follows, we address each point separately, hopefully answering and clarifying all items. If anything further can be done from our side to facilitate this process, kindly let us know.
>
> 1. For someone not already closely familiar with the signature kernel (in particular as a solution to a Goursat PDE), the mathematical derivations can be difficult to follow.
>
> Reply: Thank you for pointing this out. We agree and have revised Section 2.1 accordingly by providing additional context and references on the Goursat PDE formulation of the signature kernel. We will also streamline the exposition of Sections 2.2 and 2.3 to improve readability, in particular by adding short "road‑map" paragraphs that clearly outline the purpose of each step before presenting the detailed technical derivations.
>
> 2. To further demonstrate the usefulness of the proposed method and reinforce the claim that it is suitable for large-scale applications in different domains, it could have been interesting to show how it could allow solving a previously intractable real-world problem.
>
> Reply: We agree. In response to this, we have benchmarked $\texttt{PowerSig}$ on the kernel regression task for bitcoin pricing presented in [Salvi et al: "The Signature Kernel is the Solution of a Goursat PDE", SIAM J. Math. Data Sci., Vol. 3 (3)], where our $\texttt{PowerSig}$ achieved improved accuracy over $\texttt{KSig-PDE}$ while using only a fraction of the memory. This experiment provides an initial indication that $\texttt{PowerSig}$ enables accuracy gains in practical kernel computations on real-world datasets that can become prohibitively large for existing signature kernel methods. We are currently working to demonstrating broader applicability in the spirit of your suggestion. So far, this effort has been hindered by severe resource limitations: competing methods such as $\texttt{KSig-PDE}$ and $\texttt{KSig-RFSF}$ failed to compile or run due to excessive memory requirements, and their default configurations proved unstable (e.g., generating ill-conditioned Gram matrices) without extensive, undocumented preprocessing and hyperparameter tuning.
>
> That said, we are actively working to expand our benchmarks before the end of the rebuttal period to include additional real-world problems that have previously been out of reach for kernel-based methods. Our goal is to include at least a few such large-scale applications in the camera-ready version.
>
> 3. Could the algorithm be modified to start at any of the four corner tiles? If not, why? If yes, could you simultaneously start from every corner and process tiles in parallel towards the middle of the square?
>
> Reply: Thank you for the question and its follow-up. Indeed, the algorithm can only start at tiles where the boundary conditions of the Goursat PDE are fully specified, which in our formulation is exclusively the bottom-left corner tile. The reason is that propagating the signature kernel via the Goursat PDE on a given tile (say, $[\sigma_k, \sigma_{k+1}]\times[\tau_l,\tau_{l+1}]$) requires knowledge of its initial conditions on both associated time-front edges ($\{\sigma_k\}\times[\tau_l,\tau_{l+1}]$ and $[\sigma_k, \sigma_{k+1}]\times\{\tau_l\}$), which intersect at the tile's bottom-left corner. These initial values are only known a priori on the global boundary edges $\{0\}\times[0,1]$ and $[0,1]\times\{0\}$, where the kernel is identically $1$ by definition. These edges intersect at the bottom-left tile of the grid, making it the only valid starting point under the current scheme. To initiate propagation from a different corner, one would need to perform a tile rotation by applying a suitable change-of-variables to the time series data. We will add a remark in our revised exposition to clarify this.
>
> 4. Can you combine aspects of both your method and $\texttt{polysigkernel}$, or are they incompatible with each other?
>
> Reply: Thank you for this question. Both our method ($\texttt{PowerSig}$) and the concurrent $\texttt{polysigkernel}$ essentially target the same analytic object—the solution of the signature kernel's Goursat PDE—and yield discretizations via recursively propagated local power series expansions. They differ mainly conceptually and in their mathematical derivations: $\texttt{polysigkernel}$ constructs its localised series expansions from an asymptotic representation involving integrated Bessel functions and analytic boundary conditions, while our approach develops and employs a direct functional-analytic perspective using a Neumann series representation based on the spectral properties of the integral operator that propagates the signature kernel via its characterising Goursat PDE; we elaborate on this distinction in a newly added comment in the manuscript.
>
> Despite this methodological distinction, the resulting algorithms are structurally very similar (though not entirely, and to our advantage, as newly added empirical benchmarks indicate), sharing recursion schemes, GPU-parallelizable wavefront propagation, and similarly rapid factorial convergence of the propagated power series. The implementations of our methods are indeed compatible: one could, for instance, replace the Neumann update in our solver with $\texttt{polysigkernel}$'s Chebyshev step on selected tiles, or incorporate our low‑memory propagation update into their code. That said, given the close alignment, we believe that such combinations would offer little additional benefit. Motivated by your question, however, we see greater potential in selectively integrating finite-difference or FEM-based local solvers of the Goursat PDE (as used in $\texttt{polysigkernel}$) on particularly challenging subclusters of the tile grid. This hybrid approach could enhance robustness while maintaining memory efficiency and parallel scalability. We will include a brief discussion of this in the revised manuscript.

---

> > ### Author Response · Authors · 2025-08-08
> >
> > Since submitting our above rebuttal, we have further consolidated additional empirical illustrations of $\texttt{PowerSig}$'s practical utility on real-world benchmarks, including unmatched accuracies in kernel-based time-series prediction for bitcoin pricing resp. in a classification task from behavioural genetics, as stated resp. anticipated in our previous comments.
> >
> > Moreover, and directly addressing your earlier comment (*``To further demonstrate the usefulness of the proposed method$\ldots$''*), we can now also report an additional use case of the kind you suggested, namely one in which $\texttt{PowerSig}$'s ability to efficiently process long time series serves as a key enabler of consistent and previously intractable accuracy gains. Specifically, we considered periodic time-series data with adjustable period length, as encountered in applications such as near-periodic vibration patterns in predictive maintenance (e.g., gearboxes, turbines) or in-phase/quadrature representations of narrow-band radio signals. For these types of data, which we will also include with the camera-ready version, SVM-based downstream regressors consistently exhibited marked improvements in accuracy as the input time-series window length increased.
> >
> > This favourable accuracy trend persisted well beyond the sequence lengths that conventional or low-rank-based signature kernel methods could handle, giving $\texttt{PowerSig}$ a comparative advantage that translates into concrete, unmatched gains in downstream predictive performance and showcases a practical use case with a performance profile previously out of reach.
> >
> > We would be happy to address any remaining questions or comments you may have. Otherwise, we would once again like to thank you sincerely for your time, valuable constructive feedback, and careful consideration throughout the review process.

---

> ### Comment · Reviewer_p9wB · 2025-08-08
>
> Thank you for providing a detailed rebuttal and this follow-up, in particular regarding your efforts to improve the paper clarity and expand the evaluation to more complex scenarios. I appreciate the explanation about the boundary conditions at the other corners of the grid.

---

### Official Review · Reviewer_VWgU · 2025-07-03

**Clarity:** 3
**Significance:** 2
**Originality:** 3
**Rating:** 5
**Confidence:** 4

**Summary:**

The authors present a scalable approach for computing the signature kernel that leverages adaptively truncated local Neumann series expansions of the Goursat PDE (a hyperbolic boundary value problem) associated with the kernel. The authors show that the proposed approach is applicable to time-series with a high degree of roughness/volatility, enables massive parallelism, and achieves orders of magnitude reduction in memory requirements compared to dynamic programming and PDE based approaches.

**Questions:**

Here are some suggestions for additional numerical studies that I believe will strengthen the submission:

To make a compelling case for PowerSig, I suggest that the authors provide numerical studies on at least a few learning tasks; e.g., Multivariate UEA Datasets, https://github.com/tgcsaba/KSig  and compare against alternative approaches, Given the efficiency of PowerSig, such studies should be feasible within a reasonable computational budget. This will also provide some insights into whether the benefits of (almost) exact signature kernels translate into superior predictive performance in practice.

Numerical studies on higher-dimensional paths (say d=10, 50) to confirm that the runtime grows linearly in d and accuracy remains stable.

Numerical studies showing how P should scale as a function of roughness and sequence length to maintain a target accuracy.

Minor suggestions for clarity:

Rename the baseline explicitly in both caption and axis label, e.g. ``MAPE relative to KSig truncated-signature kernel (level 21).''

Keep the baseline constant across panels or state the new baseline explicitly in the right-hand axis label.

**Ethical Concerns:**

["NO or VERY MINOR ethics concerns only"]

**Final Justification:**

The authors have thoughtfully addressed all the points raised in my review. They have shared results from additional numerical studies  they conducted to address my concerns. This is an excellent paper and I am happy to raise my score from 4 to 5.

**Limitations:**

yes

**Quality:**

3

**Strengths And Weaknesses:**

The authors demonstrate the ability of their approach to handle time-series with over $10^6$ time-stamps on a single GPU with 24 GB RAM. This work therefore presents an impressive advance over existing methods for near-exact kernel computation over long paths.

The paper presents a linear sequence of lemmas and propositions that rigorously justify, step-by-step, the key ingredients of their computational strategy. This principled, structured presentation enhances clarity. The authors show that their approach ensures convergence when the truncation order goes to infinity.

The authors present numerical studies for a two-dimensional Brownian motion test case and compare against KSig PDE. Results are also presented for two-dimensional fractional Brownian motion paths of fixed length across decreasing Hurst indices. The results serve as an empirical demonstration of the memory reductions and runtime scalability for long sequence lengths.

From a numerical analysis viewpoint, it would have been more insightful if the authors provided theoretical results establishing bounds on the Gram matrix approximation error as a function of the truncation order. This would also provide useful insights to practioneers.

The paper does not compare against Random Fourier feature (RFF) approximations or low-rank approximations, stating their tendency to degrade in accuracy for larger time-series length or time series of higher dimension. While this is correct, in practice, accurate kernel computation is necessary but not sufficient for good generalization. Approximations such as RFF introduce their own inductive biases that can be beneficial for some datasets.

All the numerical studies presented in the paper fix the truncation order P=7. The results show error with respect to a reference KSig truncated signature kernel. For a machine learning venue, additional numerical studies on learning tasks would significantly strengthen the submission.

---

> ### Author Rebuttal · Authors · 2025-07-31
>
> Thank you very much for your time and your very valuable comments and insightful suggestions. In what follows, we address each of the points raised in the review separately, hopefully answering and clarifying any open questions or concerns. Kindly let us know if we should elaborate any further on any of the points.
>
> 1. From a numerical analysis viewpoint, it would have been more insightful if the authors provided theoretical results establishing bounds on the Gram matrix approximation error as a function of the truncation order. This would also provide useful insights to practioneers.
>
> Reply: Thank you for suggesting this. We agree that an explicit bound on the Gram matrix approximation error would be valuable, particularly for practitioners. To address this, we will add a corresponding result at the end of Section 2. This follows directly by refining our existing remarks on truncation error in Appendix B.3 into a fully specified inequality, which is straightforward: the double-sum structure of the displayed approximation error (as displayed just before (49)) admits direct control via modified Bessel tails, which can then be straightforwardly scaled from component-wise to matrix-level bounds.
>
> 2. The paper does not compare against Random Fourier feature (RFF) approximations or low-rank approximations, stating their tendency to degrade in accuracy for larger time-series length or time series of higher dimension. While this is correct, in practice, accurate kernel computation is necessary but not sufficient for good generalization. Approximations such as RFF introduce their own inductive biases that can be beneficial for some datasets.
>
> Reply: Thank you for pointing this out. We are currently implementing a comparison against low-rank kernel approximation schemes by benchmarking $\texttt{PowerSig}$ against the recent specialized low-rank and RFF-based methods introduced in [Tóth, Oberhauser, and Szabó: “Random Fourier Signature Features,” *SIAM J. Math. Data Sci.*, Vol. 7(1)]. We expect to have results within the coming days and plan to include them in the camera-ready version.
>
> On a preliminary note, to evaluate performance in a downstream learning task and to compare with RFF-based methods, we obtained preliminary test results on $\texttt{PowerSig}$, RFSF-TRP, and traditional linear/RBF kernel SVMs on the standard Eigenworms time series classification dataset for increasing time-series lengths, from within the UEA dataset that you kindly referenced. Due to the extensive preprocessing involved in such tasks, it's difficult to fully assess the potential of signature kernels on long time series out of the box at this current moment. However, to highlight practical applicability, we have obtained preliminary benchmarks on accuracy, precision, and recall across multiple truncation lengths of the Eigenworms dataset ($L = 50, 100, 200, 300,$ etc.). Over this range, we observed the accuracy of $\texttt{KSig}$ and RFSF-TRP drop from 48% to 43.5%, while the accuracy of both $\texttt{KSigPDE}$ and $\texttt{PowerSig}$ increased from 32.82% to 40.46%. We also observed that, contrary to $\texttt{PowerSig}$, RFSF-TRP begins to run out of memory at length 1000 under default settings even on an H100 GPU with 80 GB of RAM.
>
> 3. All the numerical studies presented in the paper fix the truncation order P=7. The results show error with respect to a reference KSig truncated signature kernel. For a machine learning venue, additional numerical studies on learning tasks would significantly strengthen the submission.
>
> Reply: Thank you for this suggestion, and for the reference dataset provided below. We fully agree that downstream learning tasks would be an important benchmark from a machine learning perspective. Unfortunately, practical constraints have so far limited our ability to include such experiments at a convincing scale. Specifically, we encountered severe resource limitations when attempting to reproduce kernelized learning tasks on the full-length datasets that you have kindly referenced: both $\texttt{KSig-PDE}$ and $\texttt{KSig-RFSF}$ failed to compile due to excessive memory requirements, and even the latter required significantly more memory than $\texttt{PowerSig}$ and more than our current resources could handle. Our reproducing of prior results for comparison was further complicated by undocumented yet essential and nontrivial preprocessing steps and the absence of reference hyperparameter settings on the side of competing methods, making fair or even meaningful comparisons infeasible without extensive tuning and partial reimplementation. In particular, without such tuning or preprocessing, the generated signature kernel matrices, whether from near-exact approximations or low-rank variants alike, exhibited extremely high condition numbers on the given data, which rendered standard regression and classification tasks numerically unstable or entirely infeasible.
>
> That said, we were already able to benchmark on the kernel regression dataset on bitcoin pricing showcased in [Salvi et al: "The Signature Kernel is the Solution of a Goursat PDE", SIAM J. Math. Data Sci., Vol. 3 (3)], where $\texttt{PowerSig}$ has achieved increased accuracy (1.34% MAPE) compared to $\texttt{KSigPDE}$ (1.41%) at a fraction of the memory, and we are actively working very hard to close this gap more convincingly as soon as possible and set up meaningful comparisons on selected UEA datasets and to also identify alternatives with tractable length and dimensionality where all competing methods can be run reliably. We aim to report updates on these benchmarks here as soon as possible, and certainly want to include at least a few such comparisons in the camera-ready version.
>
>    4. To make a compelling case for $\texttt{PowerSig}$, I suggest that the authors provide numerical studies on at least a few learning tasks; e.g., Multivariate UEA Datasets, https://github.com/tgcsaba/KSig and compare against alternative approaches, Given the efficiency of $\texttt{PowerSig}$, such studies should be feasible within a reasonable computational budget. This will also provide some insights into whether the benefits of (almost) exact signature kernels translate into superior predictive performance in practice.
>
> Reply: We much appreciate this suggestion and have incorporated our response into the answer to the previous point.
>
>     5. Numerical studies on higher-dimensional paths (say d=10, 50) to confirm that the runtime grows linearly in d and accuracy remains stable.
>
> Reply: Thank you for the suggestion. We have conducted the desired experiment on runtime vs time-series dimension on fixed-length (= 512) Brownian motion paths of increasing dimension, ranging from 2 to 4096, where we observed stable accuracy and confirmed the expected linear growth on runtime; indeed, we observed near perfect linear runtime growth from d=64 to d=8192, and sublinear growth outside.
>
>     6. Numerical studies showing how P should scale as a function of roughness and sequence length to maintain a target accuracy.
>
> Reply: Thank you for suggesting this. Based on the discussions in Appendix B.3, we included in our original submission a utility function to address exactly this point: estimate_required_order() (in powersig.jax.algorithm.py) takes as input time series $\(x_\mu \mid \mu = 1,\ldots, m)$, $(y_\nu \mid \nu = 1,\ldots, n)$, and a target error tolerance, and returns a recommended global truncation depth (capped at 64) for computing the cross-Gram matrix $(K_{x_\mu,y_\nu})\in\mathbb{R}^{m\times n}$. Empirically, and in line with our theoretical discussions, the required truncation depth increases with both time-series roughness and sequence length. In practice, we observed a stable operational range between orders 7 and 15, beyond which further increases brought diminishing returns; instead, rescaling the time series was generally a more effective means to restore numerical stability. We will add a corresponding remark in the main text and explicitly refer to this utility function.

---

> > ### Comment · Reviewer_VWgU · 2025-08-04
> >
> > Thank you for your response which thoughtfully addresses all the points raised in my review. I appreciate the substantial effort by the authors to conduct additional numerical studies.
> >
> > (1) The planned addition of theoretical bounds on the Gram matrix approximation error in Section 2 will significantly strengthen the theoretical contribution.
> >
> > (2) The preliminary results comparing against baseline methods and the planned inclusion of low-rank and RFF kernel approximation benchmarks directly address my concerns about empirical comparisons. I understand the practical challenges arising from compute resource constraints when addressing downstream learning tasks on the full UEA datasets. The  preliminary results you have shared on the Eigenworms dataset and the bitcoin pricing regression task provide valuable initial evidence of PowerSig's practical utility.
> >
> > (3) Thank you - your response satisfactorily addresses my comments about scalability studies.
> >
> > Given the substantive response and the planned revisions, I am happy to increase my rating from 4 to 5. This paper represents a solid theoretical and computational advance in signature kernel methods. All the best.

---

> > > ### Author Response · Authors · 2025-08-08
> > >
> > > Thank you very much for your reply and for acknowledging our rebuttal. We confirm that all of the aforementioned revisions will be fully incorporated into the camera-ready version.
> > >
> > > In addition to confirming and further consolidating the previously mentioned empirical illustrations of $\texttt{PowerSig}$'s practical utility on real-world benchmarks against both classical and low-rank or RFF-based signature kernel approximations, we can now also report an additional downstream application, detailed in our comment below, which further demonstrates how $\texttt{PowerSig}$'s ability to efficiently process long time series enables unprecedented computational feasibility that directly translates into clear and consistent gains in predictive performance, highlighting another previously unachievable performance profile in applications.
> > >
> > > We remain happy to address any remaining questions or comments. Otherwise, we thank you once again for your time, your highly valuable and constructive feedback, and your consideration. All the best.

---

### Note · Authors · 2025-08-16

We thank all reviewers and the AC for their time, constructive feedback, and engagement. We have worked to address all points raised in the reviews and will incorporate the corresponding changes into the camera-ready version.

In our rebuttal, we strengthened the theory (adding explicit Gram-matrix error bounds), clarified the exposition, and expanded experiments to include real-world and additional large-scale settings, demonstrating that $\texttt{PowerSig}$ delivers clear and consistent theoretical and practical performance advantages over conventional and low-rank-based signature kernel methods in these contexts. We also highlighted the markedly different and more concise theoretical underpinnings of $\texttt{PowerSig}$ compared to the concurrent $\texttt{polysigkernel}$ (appearing on arXiv within two weeks of our own submission) and benchmarked directly against it, demonstrating consistent and significant speed and memory advantages.

To the best of our knowledge, $\texttt{PowerSig}$ is the only method to jointly achieve near-exactness with memory scaling linearly in sequence length, demonstrated scalability to million-point time series, and robustness under extreme roughness, while also showing consistently superior performance in real-world downstream applications on long time series that were previously computationally infeasible for signature kernels.

---

### Decision · Program_Chairs · 2025-09-17

**Decision:**

Accept (poster)

**Comment:**

This manuscript proposes a method for efficiently working with the signature kernel, particularly when the underlying time-series data is long or high-dimensional. A key technical innovation is the use of local power series expansions (with appropriate surrounding machinery).

The reviewers all highlight the efficacy of the scheme; it is interesting and clearly effective. The numerical experiments clearly highlight this. Moreover, in the manuscript it is accompanied by theory well matched to its practical use.

Some minor weaknesses are in the presentation (e.g., being quite dependent on a lot of "prior" information) and, initially, the scope of some experiments. The latter was addressed during the rebuttal; the former may be somewhat intrinsic.